# Human iPSC-derived mature microglia retain their identity and functionally integrate in the chimeric mouse brain

Ranjie Xu[1], Xiaoxi Li[1,2], Andrew J. Boreland[1,3], Anthony Posyton[1], Kelvin Kwan [1], Ronald P. Hart [1] & Peng Jiang [1✉]

Microglia, the brain-resident macrophages, exhibit highly dynamic functions in neurodevelopment and neurodegeneration. Human microglia possess unique features as compared to mouse microglia, but our understanding of human microglial functions is largely limited by an inability to obtain human microglia under homeostatic states. Here, we develop a human pluripotent stem cell (hPSC)-based microglial chimeric mouse brain model by transplanting hPSC-derived primitive macrophage progenitors into neonatal mouse brains. Single-cell RNA-sequencing of the microglial chimeric mouse brains reveals that xenografted hPSC-derived microglia largely retain human microglial identity, as they exhibit signature gene expression patterns consistent with physiological human microglia and recapitulate heterogeneity of adult human microglia. Importantly, the engrafted hPSC-derived microglia exhibit dynamic response to cuprizone-induced demyelination and species-specific transcriptomic differences in the expression of neurological disease-risk genes in microglia. This model will serve as a tool to study the role of human microglia in brain development and degeneration.

[1] Department of Cell Biology and Neuroscience, Rutgers University, Piscataway, NJ 08854, USA. [2] Department of Immunology, Nanjing Medical University, Nanjing, China. [3] Graduate Program in Molecular Biosciences, Rutgers University, Piscataway, NJ 08854, USA. ✉email: peng.jiang@rutgers.edu

As the resident macrophages of the central nervous system (CNS), microglia play critical roles in maintenance of CNS homeostasis and the regulation of a broad range of neuronal responses[1]. Recent studies indicate that dysfunction of microglia contributes to neurodevelopmental and neurodegenerative diseases, including Alzheimer's disease (AD)[2,3]. Moreover, genome-wide association studies have shown that many neurological disease-risk genes, particularly neurodegenerative diseases, are highly and sometimes exclusively expressed by microglia[4]. These observations provide a compelling incentive to investigate the role of microglia in models of abnormal brain development and neurodegeneration. Most studies of microglia largely rely on rodent microglia. However, there is increasing evidence that rodent microglia are not able to faithfully mirror the biology of human microglia[5]. In particular, recent transcriptomic studies have clearly demonstrated that a number of immune genes, not identified as part of the mouse microglial signature, were abundantly expressed in human microglia[6,7]. Moreover, a limited overlap was observed in microglial genes regulated during aging and neurodegeneration between mice and humans, indicating that human and mouse microglia age differently under normal and diseased conditions[6,8]. These findings argue for the development of species-specific research tools to investigate microglial functions in human brain development, aging, and neurodegeneration.

Functional human brain tissue is scarcely available. In addition, given the considerable sensitivity of microglia to environmental changes[7], the properties of available human microglia isolated from surgically resected brain tissue may vary significantly, due to different disease states of the patients and the multistep procedures used for microglial purification. In order to study human microglia in a relatively homeostatic state, many scientists have turned to human pluripotent stem cells (hPSCs). Recent advances in stem cell technology have led to the efficient generation of microglia from hPSCs[9–13], providing an unlimited source of human microglia to study their function. However, when cultured alone or co-cultured with neurons and astrocytes in 2-dimensional (2D) or 3D organoid/spheroid culture systems, these hPSC-derived microglia best resemble fetal or early postnatal human microglia. This is indicated by much lower expression of key microglial molecules such as TREM2, TMEM119, and P2RY12 in the hPSC-derived microglia, as compared with microglia derived from adult human brain tissue[11,13,14]. Thus, even with these in vitro models, it has been challenging to advance understanding of human microglial function in adult ages or in neurodegeneration during aging.

Recent studies from us[15,16] and others[17,18] have demonstrated that neonatally engrafted human neural or macroglial (oligodendroglial and astroglial) progenitor cells can largely repopulate and functionally integrate into the adult host rodent brain or spinal cord, generating widespread chimerism. This human-mouse chimeric approach provides unique opportunities for studying the pathophysiology of the human cells within an intact brain. In this study, we develop a hPSC microglial chimeric mouse brain model, by transplanting hPSC-derived microglia into neonatal mouse brains. The engrafted hPSC-derived microglia can proliferate, migrate, and widely disperse in the brain. We hypothesize that the limited functional maturation of hPSC-derived microglia in in vitro models is primarily caused by the fact that those microglia are maintained in an environment that lacks the complex cell–cell/cell–matrix interactions existing in an in vivo brain environment[7]. To test this hypothesis, we employ single-cell RNA sequencing (scRNA-seq) and super-resolution confocal imaging to examine the identity and function of hPSC-derived microglia developed for six months in the mouse brain under both homeostatic and toxin-induced demyelination conditions.

## Results

**Generation of hPSC microglial chimeric mouse brains.** Microglia originate from yolk sac erythromyeloid progenitors (EMPs) during primitive hematopoiesis. EMPs further develop to primitive macrophage progenitors (PMPs) that migrate into the developing neural tube and become microglia with ramified processes within the CNS environment[1]. We first derived PMPs from hPSCs, including one human induced pluripotent stem cell (hiPSC) line and one human embryonic stem cell (hESC) line, using a published protocol[13]. We found that at 2–3 weeks after plating embryoid bodies (EBs), hPSC-derived PMPs emerged into the supernatant and were continuously produced for more than 3 months. The cumulative yield of PMPs was around 40-fold higher than the number of input hPSCs[11,13] (Fig. 1a). PMPs are produced in a Myb-independent manner that closely recapitulated primitive hematopoiesis[1,13,19]. We confirmed the identity of these hPSC-derived PMPs by staining with CD235, a marker for YS primitive hematopoietic progenitors[20], and CD43, a marker for hematopoietic progenitor-like cells[20]. As shown in Fig. 1b, over 95% of the hPSC-derived PMPs expressed both markers. Moreover, the human PMPs are highly proliferative as indicated by Ki67 staining (Fig. 1b). Using this method, we routinely obtain ample numbers of hPSC-derived PMPs with high purity as required for cell transplantation experiments.

We engrafted hPSC-derived PMPs into the brains of postnatal day 0 (P0) immunodeficient mice that are Rag2/IL2rγ-deficient and also express the human forms of colony-stimulating factor 1 (CSF1), which facilitates the survival of xenografted human myeloid cells and other leukocytes[21]. We deposited cells into the white matter overlying the hippocampus and sites within the hippocampal formation (Fig. 1c). In order to visualize the distribution of donor-derived microglia, at 6 months post transplantation, we stained the mouse brain sections with human-specific antibody recognizing TMEM119 (hTMEM119). TMEM119 is a marker that is only expressed by microglia, but not other macrophages[22]. We found that the donor-derived hTMEM119+ microglia widely disperse in the brain (Fig. 1d). As early as 3 weeks post transplantation, donor-derived microglia had already migrated along corpus callosum and passed through the rostral migratory stream to the olfactory bulb (Fig. 1e). At 6 months post transplantation, human microglia widely dispersed in multiple brain regions, including olfactory bulb, hippocampus, and cerebral cortex, and exhibited a highly ramified morphology (Fig. 1f, g). Frequently, we also observed clusters of human microglia in the cerebellum (Fig. 1h), which might be a result from the strong ability of immune cells trafficking along blood vessels and/or the choroid plexus[23]. Similar to previous studies[16–18], we assessed the engraftment efficiency and degree of chimerization by quantifying the percentage of hTMEM119+ cells among total DAPI+ cells in the forebrain in sagittal brain sections covering regions from 0.3 to 2.4 mm lateral to midline and found that about 8% of the total cells were human microglia in the 6-month-old mouse brains (Fig. 1d, l). As shown by individual data points overlaid on the bar graphs (Fig. 1l), there were variations in chimerization among animals. In addition to the images showing distribution of donor-derived cells in Fig. 1d, we thus also included tile scan images collected from another chimeric mouse brain that represents the lower level of chimerization (supplementary Fig. 1A). In the developing brain, microglia are known to use white matter tracts as guiding structures for migration and that they enter different brain regions[24]. In order to examine the migration pattern of our transplanted cells, we deposited PMPs into different sites, the lateral ventricles of P0 mice. As early as 3 weeks post transplantation, we found that the majority of donor-derived cells migrated along the anterior

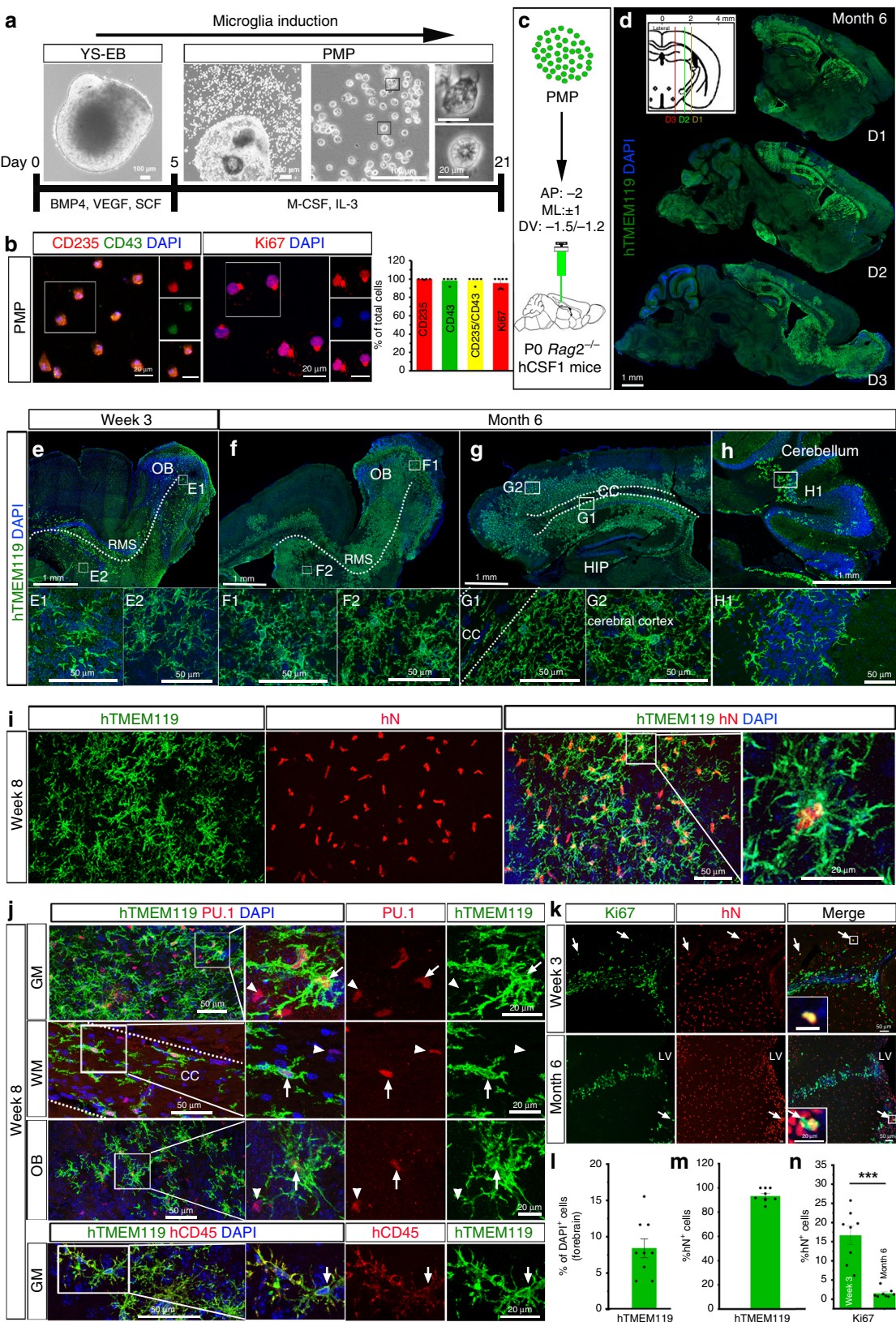

corpus callosum, rostral migration stream, and then entered the olfactory bulb. Moreover, some of those cells of migrated posteriorly along the corpus callosum (Supplementary Fig. 1B), suggesting that engrafted cells likely used corpus callosum to migrate to various brain regions. These results demonstrate that

hPSC-derived PMPs survive in mouse brain and that they migrate to a variety of structures.

To examine whether transplanted hPSC-derived PMPs efficiently differentiated to microglia in the mouse brain, we double-stained brain sections for both human nuclei (hN) and

**Fig. 1 Generation of hPSC microglial chimeric mouse brains. a** A schematic procedure for generating primitive macrophage progenitor (PMP) from hiPSCs or hESCs-derived yolk sac embryoid bodies (YS-EB). Insets: representative bright-field images at different stages. Scale bars represent 100, 200, and 20 μm as indicated in the images. **b** Representative images and quantification of CD235[+], CD43[+], CD235[+]/CD43[+], and Ki67[+] cells in PMP. Quantification of pooled data from one hiPSC line and one hESC line. The experiments are repeated for five or seven times ($n = 5$ for CD235[+], CD43[+], and CD235[+]/CD43[+], $n = 7$ for Ki67[+]) and for each experiment, the two stem cell lines are used. Data are presented as mean ± s.e.m. Scale bars: 20 μm in the original and enlarged images. **c** A schematic diagram showing that hPSC-derived PMP are engrafted into the brains of P0 $rag2^{-/-}$ hCSF1 mice. **d** Representative images from sagittal brain sections showing the wide distribution of xenografted hPSC-derived microglia at 6 months post transplantation. Anti-human-specific TMEM119 (hTMEM119) selectively labels xenografted hPSC-derived microglia. Scale bar: 1 mm. **e–h** Representative images from sagittal brain sections showing the distribution of hTMEM119[+] xenografted hPSC-derived microglia at 3 weeks and 6 months post transplantation in different brain regions. OB olfactory bulb, RMS rostral migratory stream, CC corpus callosum, HIP hippocampus. Scale bars: 1 mm or 50 μm in the original or enlarged images, respectively. **i** Representative images of hTMEM119[+] cells among the total donor-derived hN[+] cells in gray matter at 8 weeks post transplantation. Scale bars, 50 μm or 20 μm in the original or enlarged images, respectively. **j** Representative images of PU.1- and human-specific CD45 (hCD45)-expressing cells in the donor-derived hTMEM119[+] cells in different brain regions at 8 weeks post transplantation. Scale bars, 50 μm or 20 μm in the original or enlarged images, respectively. **k** Representative images of Ki67[+] cells among the total donor-derived hN[+] cells at 3 weeks and 6 months post transplantation. Scale bars, 50 μm or 20 μm in the original or enlarged images, respectively. **l** Quantification of the percentage of hTMEM119[+] cells in total DAPI[+] cells in the forebrain at 6 months post transplantation ($n = 9$ mice). The data are pooled from the mice received transplantation of microglia derived from both hESCs and hiPSCs. Data are presented as mean ± s.e.m. **m** Quantification of the percentage of hTMEM119[+] cells in total hN[+] cells ($n = 7$ mice). Data are presented as mean ± s.e.m. **n** Quantification of Ki67[+] cells among the total donor-derived hN[+] cells at 3 weeks or 6 months post transplantation ($n = 8$ mice for each time point). The data are pooled from the mice that received transplantation of microglia derived from both hESCs and hiPSCs. Student's $t$ test (two-sided). ***$P < 0.001$. Data are presented as mean ± s.e.m. Source data are provided as a Source Data file.

hTMEM119. As early as 8 weeks post transplantation, the vast majority of hN[+] donor-derived cells (91.7 ± 2.1%, $n = 7$) were positive for hTMEM119 (Fig. 1i, m), indicating the robust and efficient differentiation of hPSC-derived PMPs into microglia. Moreover, the vast majority of the donor-derived cells expressed PU.1, a transcription factor that is necessary for microglial differentiation and maintenance[25], and were positive for human-specific CD45 (hCD45), which is expressed by all nucleated hematopoietic cells[26] (Fig. 1j). Similarly, at 6 months post transplantation, the vast majority of donor-derived cells expressed hTMEM119 (Supplementary Fig. 2A) and P2RY12 (93.4 ± 3.8%, $n = 7$; Supplementary Fig. 2B, C). hTMEM119 and P2RY12 was not expressed in PMP cultures (Supplementary Fig. 2D). Moreover, we examined the distribution of human donor cells in border regions, including choroid plexus, meninges, and perivascular spaces. We found that most of the hTMEM119[−]/hN[+] donor-derived cells were seen in those border regions. Furthermore, we co-stained hTMEM119 with CD163, an established marker for non-microglial CNS myeloid cells[27]. In choroid plexus, we found that some of the hTMEM119[−] cells co-expressed CD163, suggesting that these transplanted cells differentiated into choroid plexus macrophage, but not microglia (Supplementary Fig. 3A). In order to better visualize meninges and perivascular space, we triple-stained hN and CD163 with laminin, a marker that has been commonly used to visualize vascular structures in the mammalian brain[28]. There was also a small number of hN[+] and CD163[+] co-expressing cells in these regions, indicating that the transplanted cells differentiated into meningeal macrophage and perivascular macrophages (Supplementary Fig. 3B, C). There was a possibility that some transplanted cells might remain as progenitors and maintain their hematopoietic progenitor-like cell identity. As shown in Supplementary Fig. 3D, we found that a small population of hN[+] cells expressed CD235 in the regions close to the lateral ventricles. Overall, these results demonstrate that the vast majority of engrated hiPSC-derived PMPs differentiate into hTMEM119[+] microglia, with a relatively small number giving rise to other of hTMEM119[−] CNS myeloid cells in a brain context-dependent manner or remaining as progenitors.

We next assessed the proliferation of engrafted cells by staining the proliferative marker Ki67. As shown in Fig. 1k, n, at 3 weeks post transplantation, about 17% of hN[+] transplanted cells expressed Ki67, indicating that these cells were capable of

proliferating in the mouse brain. At 6 months post transplantation, the percentage of proliferating cells dramatically decreased and <2% of total engrafted cells were Ki67 positive. These Ki67[+] proliferating human cells mainly localized in the subventricular zone, the walls along lateral ventricles, corpus callosum, and olfactory bulb (Fig. 1k and Supplementary Fig. 3E). We also examined the proliferation of mouse host microglia at different brain regions at 3 weeks post transplantation. We only found a very small number of Ki67[+] mouse microglia in the subventricular zone (Supplementary Fig. 3F). Taken together, these findings demonstrate that engrafted hPSC-derive PMPs differentiate to microglia, generating a mouse brain with a high degree of human microglial chimerism in the forebrain.

**Characterization of engrafted human microglia.** Compared with 3 weeks post transplantation, hPSCs-derived microglia appeared to exhibit more complex processes at 6 months post transplantation (Fig. 1e, f). Moreover, even at the same stage, hPSC-derived microglia in the cerebral cortex seemed to exhibit much more complex morphology, compared with the hPSCs-derived microglia in the corpus callosum and cerebellum (Fig. 1g, h and hTMEM119 and Iba1 staining in Supplementary Figs. 2A and 4, respectively). In the corpus callosum, hPSC-derived microglia had fewer branches that aligned with axons; and in the cerebral cortex, the microglia exhibited more complex and ramified processes (Supplementary Figs. 2A and 4), similar to observations from previous studies[24]. This prompted us to further examine the morphological and functional changes of the hPSC-derived microglia along with the development of the mouse brain, particularly in cerebral cortex. Previous studies have shown that there are no changes in microglial number, cytokine levels, and gene expression profiles between wild-type and $Rag2^{-/-}$ mice[29]. Building upon that, we also compared the differences between xenografted hPSC-derived microglia versus host mouse microglia. We double-stained the brain sections with human and mouse specific TMEM119 (hTMEM119 and mTMEM119, respectively) antibodies to distinguish hPSC-derived microglia and mouse host microglia (Fig. 2a, b). As shown in Fig. 2a, in 6-month-old mice, both hPSCs-derived microglia and mouse microglia were seen in the cerebral cortex and hippocampus. Notably, we observed that mouse microglia mainly resided in distal regions in the cerebral cortex and hippocampus. In particular, in the corpus callosum,

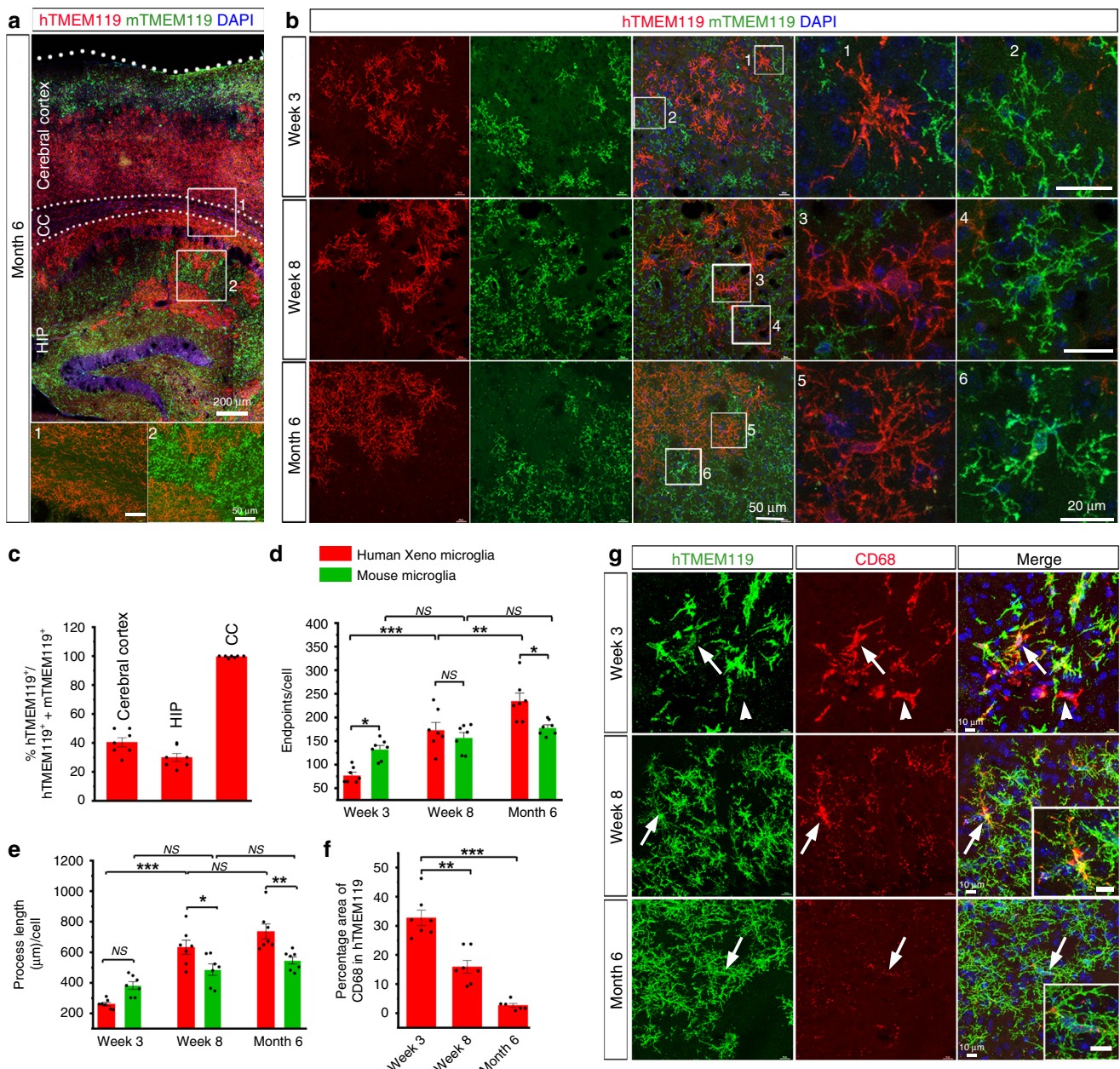

**Fig. 2 Human PSC-derived microglia undergo morphological maturation. a** Representatives of hTMEM119[+] hPSC-derived microglia and mTMEM119[+] mouse microglia in the cerebral cortex, corpus callosum (CC) and hippocampus (HIP) in 6-month-old mice. Scale bars: 200m and 50 μm in the original and enlarged images, respectively. **b** Representatives of hTMEM119[+] hPSC-derived microglia and mTMEM119[+] mouse microglia in the cerebral cortex. Scale bars: 50 and 20 μm in the original and enlarged images, respectively. **c** Quantification of the percentage of hTMEM119[+] cells in total microglia (hTMEM119[+] plus mTMEM119[+]) in the cerebral cortex, hippocampus (HIP), corpus callosum (CC) in 6-month-old chimeric mice ($n = 7$ mice for each time point). The data are pooled from the mice received transplantation of microglia derived from both hESCs and hiPSCs. Data are presented as mean ± s.e.m. **d, e** Quantification of endpoint numbers and total process length of mouse and hPSC-derived microglia based on mTMEM119 or hTMEM119 staining, respectively, from gray matter at the three time points ($n = 7$ mice for each time point). The data are pooled from the mice received transplantation of microglia derived from both hESCs and hiPSCs. Two-way ANOVA is used to compare the endpoints and process length between human and mouse microglia and one-way ANOVA is used for the comparison within mouse or human microglia. *$P < 0.05$, **$P < 0.01$, ***$P < 0.001$, and NS no significance. Data are presented as mean ± s.e.m. **f** Quantification of the percentage of CD68[+] area in hTMEM119[+] area from cerebral cortex in 3-, 8- and 6-month-old chimeric mice ($n = 7$ mice for each time point). The data are pooled from the mice received transplantation of microglia derived from both hESCs and hiPSCs. One-way ANOVA test, **$P < 0.01$, ***$P < 0.001$. Data are presented as mean ± s.e.m. Source data are provided as a Source Data file. **g** Representatives of CD68- and hTMEM119-exprssing cells in the cerebral cortex. Scale bars: 10 or 2 μm in the original or enlarged images, respectively.

mouse microglia were rarely seen, and the vast majority of microglia were hPSC-derived microglia (Fig. 2c), indicating that hPSC-derived microglia replaced the host mouse microglia. In the cerebral cortex, hTMEM119[+] hPSC-derived microglia exhibited much more complex processes at 8 weeks and 6 months post transplantation than those cells at 3 weeks post transplantation, as indicated by the increased number of endpoints (Fig. 2d). The total length of processes of hPSC-derived microglia also significantly increased from week 3 to week 8 and month 6 (Fig. 2e), suggesting the gradual maturation of hPSC-derived microglia in

mouse brain. We further examined the morphological differences between hPSC-derived microglia versus mouse microglia at the same time points after transplantation. In the cerebral cortex, at 3 weeks post transplantation, compared with hPSC-derived microglia, mouse microglia showed a significantly higher number of endpoints and a slight trend of longer processes (Fig. 2d, e). However, at 8 weeks post transplantation, there was no significant difference in endpoint number and process length between hPSC-derived microglia and mouse microglia (Fig. 2d, e). Interestingly, at 6 months post transplantation, hPSC-derived microglia exhibited a significantly higher number of endpoints and longer process length than mouse microglia. Since microglial morphology is inextricably linked to their phagocytic functions[30], we examined the expression of CD68, a marker for the phagolysosome[31]. In the cerebral cortex, CD68 was expressed in some of the hPSC-derived microglia at 3 weeks post transplantation and its expression dramatically decreased from 8 weeks to 6 months post transplantation (Fig. 2f, g). We observed some hTMEM119$^-$/CD68$^+$ cells at 3 weeks and nearly no hTMEM119$^-$/CD68$^+$ cells at 6 months post transplantation (Fig. 2g), suggesting that nearly no host mouse microglia expressed CD68 at 6 months post transplantation. Taken together, hPSC-derived microglia show variable morphologies in a spatiotemporal manner and morphologically differ from the host mouse microglia.

Microglia have been shown to shape synapse formation by pruning synapses and to maintain oligodendroglial homeostasis by phagocytizing oligodendroglial cells[32,33]. We therefore investigated whether hPSCs-derived microglia are functional in the mouse brain. To examine synaptic pruning function, we employed a super-resolution imaging technique to visualize synapse engulfment by hPSCs-derived microglia. We triple-stained hCD45 with both a postsynaptic marker PSD95 and a presynaptic marker synapsin I. The 3D reconstruction images show that PSD95$^+$ and synapsin I$^+$ puncta are colocalized within hCD45$^+$ processes, indicating that these synaptic proteins are phagocytosed by hPSCs-derived microglia at 8 weeks post transplantation in gray matter (Fig. 3a). In addition, we also validated the specificity of PDS95 puncta staining by incubating brain sections with the PSD95 antibody together with a PSD95 peptide. We barely detected any PSD95$^+$ puncta signal after the incubation in the presence of PSD95 peptide (Supplementary Fig. 5A). We also triple-stained hTMEM119 and PSD95 with CD68. As shown in Fig. 3b and Supplementary Fig. 5B, C, PSD95$^+$ puncta are localized within CD68$^+$ phagolysosomes in hTMEM119$^+$ hPSCs-derived microglia, further indicating their synaptic pruning function. Of note, this engulfment of synaptic materials was observed from 3 weeks to 6 months post transplantation, with a peak at 8 weeks post transplantation (Fig. 3c). Very few mouse microglia were found to engulf synaptic proteins at 8 weeks post transplantation (Supplementary Fig. 6A). To examine the function of phagocytosing oligodendroglia, we double-stained hCD45 with PDGFRα, a marker for oligodendroglial progenitor cell. We observed that hCD45$^+$ human microglia were able to engulf PDGFRα$^+$ oligodendroglia at 3 weeks post transplantation in the corpus callosum (Fig. 3d). We also double-stained hCD45 with the oligodendroglial marker OLIG2 and similarly found that hPSCs-derived microglia in white matter engulfed OLIG2$^+$ oligodendroglia at 3 weeks post transplantation (Supplementary Fig. 6B). In addition, we detected that a small population of mouse microglia engulfed OLIG2$^+$ oligodendroglia in the corpus callosum at 3 weeks post transplantation (Supplementary Fig. 6C).

Microglia, together with endothelial cells, pericytes and astrocytes, form the functional blood–brain barrier[34,35]. We double-stained brain sections with hCD45 and laminin. We found that hPSC-derived microglia clustered around and were closely affiliated with laminin$^+$ blood vessels in both gray matter

and white matter across different brain regions including the olfactory bulb (Fig. 3e and Supplementary Fig. 6D). In addition, we also found that mouse microglia similarly had close contact with blood vessels in the cerebral cortex, corpus callosum, and olfactory bulb (Supplementary Fig. 6E). Altogether, these results demonstrate hPSCs-derived are functional in the mouse brain under homeostatic conditions. The human microglia and host mouse microglia exhibited similar microglial functions, including synaptic pruning, phagocytosis of oligodendroglia, and having contact with blood vessels.

**ScRNA-seq of hiPSC microglial chimeric mouse brain.** Homeostatic human microglia at adult stages are difficult to obtain, because microglia are highly sensitive to environmental changes and microglia derived from adult human brain tissue-derived are usually purified through multistep procedures that can change their biological properties significantly[7]. In addition, microglia derived from hPSCs using all current differentiation protocols largely resemble fetal or early postnatal human microglia[11,13,14]. We hypothesize that hPSC microglial chimeric mice may provide a unique opportunity to study biological properties of adult human microglia, because the engrafted hPSC-derived microglia are likely to exhibit an expedited maturation process promoted by the maturing environment in the mouse brain. To test this hypothesis, we examined transcriptomic profiles of hiPSC-derived microglia, developed in the in vivo homeostatic mouse environment, using scRNA-seq. We collected brain regions where engrafted hiPSCs-derived microglia preferentially dispersed, including the cerebral cortex, hippocampus, corpus callosum, and olfactory bulb, from 6-month-old chimeric mouse brain for scRNA-seq. A previous study has demonstrated that within hours during which microglia are isolated from the brain environment and transferred to culture conditions, microglia undergo significant changes in gene expression[7]. To capture observed expression patterns as close to the "in vivo" patterns as possible, we chose to omit a FACS sorting step since it would have added substantial processing time. Owing to the wide distribution and high abundance of hiPSC-derived microglia in those brain regions, we were able to capture ample numbers of hiPSC-derived microglia for scRNA-seq even without enrichment by FACS sorting. After brain tissue dissociation with papain and centrifugation to remove debris and myelin, single-cell suspensions were directly subjected to droplet-based 10X Genomic RNA-seq isolation (Fig. 4a). Using stringent criteria, 29,974 cells passed the quality control evaluation (with about 10,000–15,000 reads/cell) from four animals for downstream analysis (Supplementary Fig. 7A).

We performed dimensionality reduction and clustering using a principal component analysis (PCA)-based approach. Using t-distributed stochastic neighbor embedding to visualize cell clustering, we identified 11 clusters, including a cluster of xenografted hiPSC-derived microglia, which we named Xeno MG (Fig. 4b). This clustering pattern was consistently and independently obtained in all four animals, indicating the reproducibility of the sequencing and clustering procedures (Supplementary Fig. 7B). We defined each cluster based on the expression of enriched genes (Supplementary Data 1) that could be recognized as markers for specific cell types or are reported to be abundantly expressed in specific cell types (Fig. 4c and Supplementary Fig. 7C). The clusters included ten mouse cell types: astrocytes (SCL6A11 (ref. [36]), NTSR2 (ref. [37])), oligodendrocytes (CLDN11 (ref. [38]), CNP[39]), oligodendrocyte progenitor cells (OPC; PDGFRα, OLIG2)[40], excitatory neurons (SYT1 (ref. [41]), SNAP25 (ref. [42])), neuronal precursors (SOX11 (ref. [43]), STMN2 (ref. [44])), vascular cells (MYL9 (ref. [45]), MGP[46]),

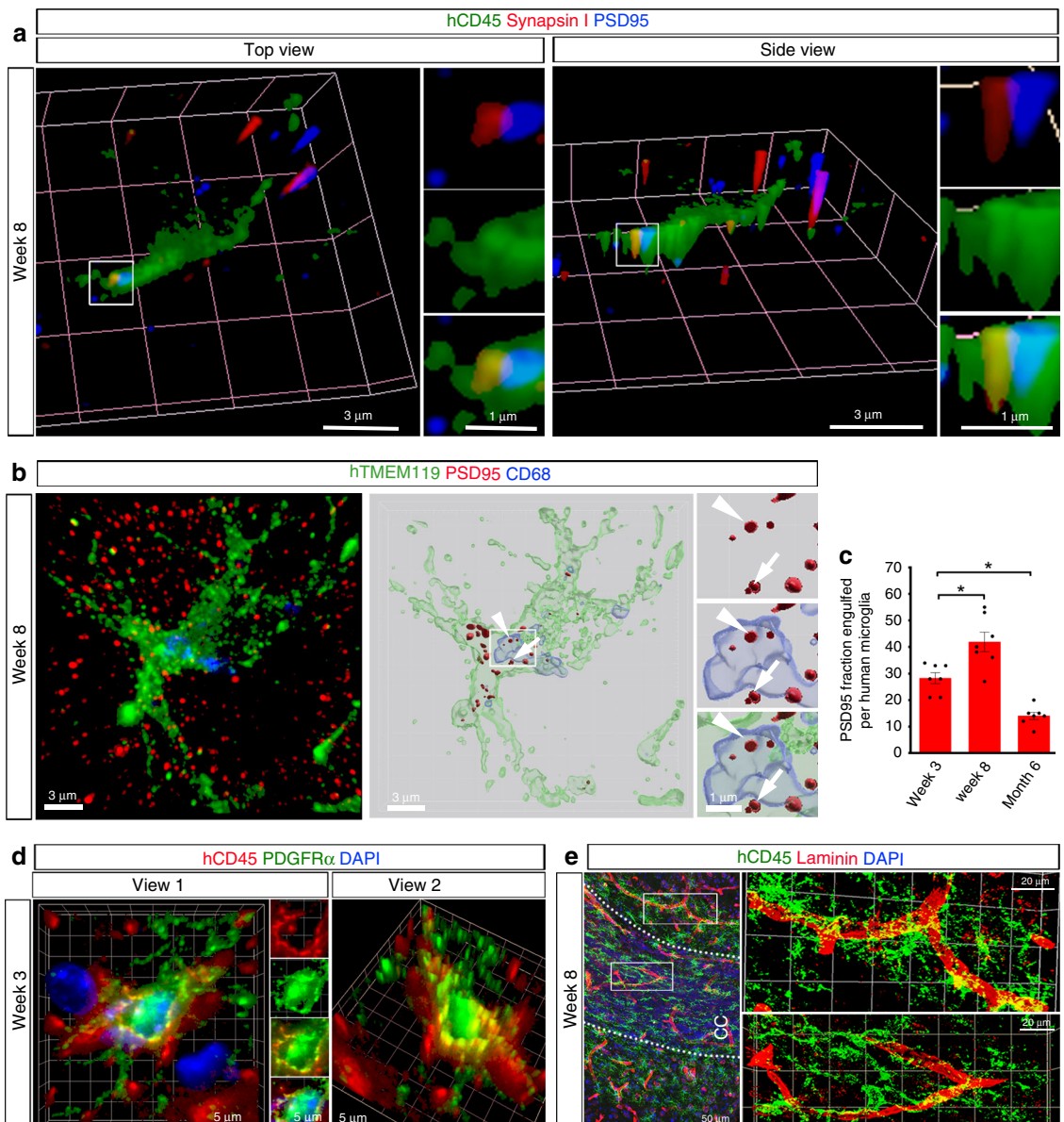

**Fig. 3 Human PSC-derived microglia are functional in the mouse brain. a** Representative 3D reconstruction of super-resolution images showing hCD45+ donor-derived microglia engulf synaptic proteins, synapsin I, and PSD95 in gray matter at 8 weeks post transplantation. Scale bars, 3 and 1 μm in the original or enlarged images, respectively. **b** Representative images showing colocalization of hTMEM119 (green), PSD95 (red), and CD68 (blue) staining in gray matter at 8 weeks post transplantation. Left, a raw fluorescent super-resolution image. Right, a 3D-surface rendered image from the same raw image. Arrows and arrowheads depict PSD95 (red) engulfed within microglia (green) or within lysosome (blue), respectively. Scale bars, 3 and 1 μm in the original or enlarged images, respectively. **c** Quantification of PSD95+ fraction engulfed per microglia from cerebral cortex at the three time points ($n = 7$ mice for each time point). The data are pooled from the mice received transplantation of microglia derived from both hESCs and hiPSCs One-way ANOVA test, *$P < 0.05$. Data are presented as mean ± s.e.m. Source data are provided as a Source Data file. **d** Representative 3D reconstruction of super-resolution images showing that hCD45+ hPSC-derived microglia phagocytize PDGFRα+ oligodendroglial cells in the corpus callosum at 3 weeks post transplantation. Scale bars, 5 μm. **e** Representative images showing the interaction between laminin+ blood vessels and hCD45+ human microglia in gray matter and white matter at 8 weeks post transplantation. Scale bars, 50 and 20 μm in the original or enlarged images, respectively.

choroid cells (*LCN2* (ref. [47]), *1500015O10Rik*[48]), endothelial cells (*ITM2A*[49], *FLT1* (ref. [50])), GABAergic neuron (*NPY*[51], *NR2F2* (ref. [52])) and mouse microglia (P2RY12, *C1QA*, and *CX3CR1*). The only human cell cluster, labeled Xeno MG, similarly expressed the microglial markers P2RY12, *C1QA*, and *CX3CR1*, and accounted for about 7% of total cells in our selected dissected regions (Supplementary Fig. 7D). Of note, a cross-correlation analysis of clustered cell types showed that Xeno MG had a highest correlation coefficient value (0.776) with mouse microglia, consistent with a microglial identity of the engrafted human cells

(Supplementary Fig. 7E). Furthermore, the expression of a set of canonical microglial genes (*C1QA*, *CX3CR*, *TREM2*, *CSF1R*, and P2RY12) was only detected in Xeno MG and mouse microglia clusters (Fig. 4d). Moreover, we performed bulk RNA-seq to analyze the pre-engraftment hiPSC-derived PMPs. We compared transcriptomic profiles between PMPs and Xeno MG from 6-month-old chimeric mice. Notably, as shown in Fig. 4e, compared with PMPs, Xeno MG highly expressed microglial identity markers, such as *TMEM119*, P2RY12, *SALL1*, and *OLFML3*, which were barely detected in PMPs. On the other

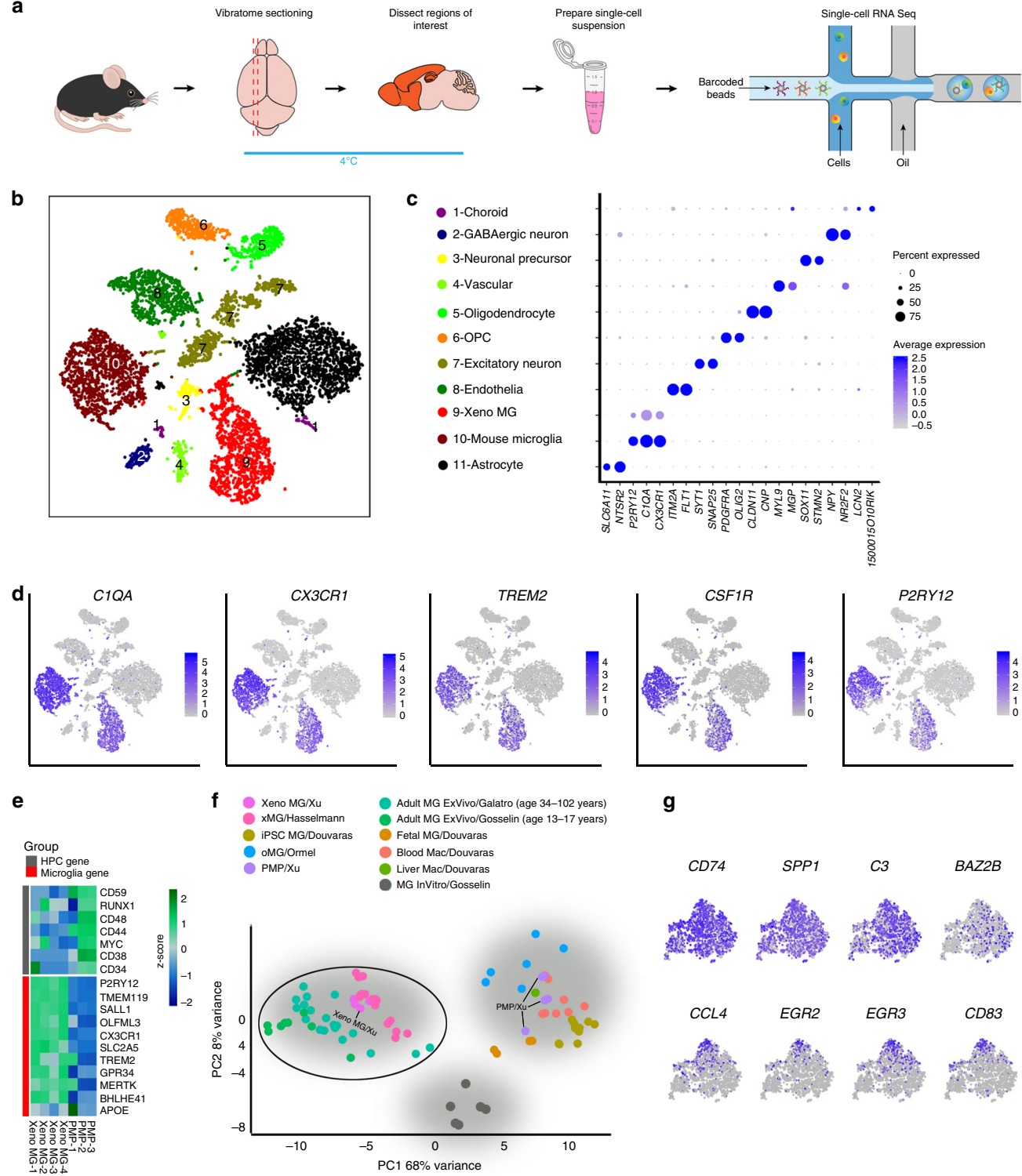

hand, the expression of markers for hematopoietic progenitor cells, such as *CD59*, *CD44*, and *CD38*, was low in Xeno MG but much higher in PMPs. Furthermore, we compared the transcriptomic profile of Xeno MG with a published dataset generated from human brain tissue-derived human microglia[6]. A significant correlation was observed between Xeno MG and the published dataset[6] (Supplementary Fig. 8A), further confirming the human microglial identity of the engrafted human cells. As shown in Figs. 1 and 2, the highly ramified morphology of hiPSC-derived

microglia suggest that they stay at a homeostatic state. We examined expression of several pro-inflammatory cytokines to assess the impact of the tissue preparation procedures on the microglial state. Consistently, we found very minor expression of acute pro-inflammatory cytokines such as IL-1β, IL-1α, and TNF-α (Supplementary Fig. 8B). In contrast, the pro-inflammatory cytokine, IL-6 and an anti-inflammatory cytokine, IL-10, were nearly undetectable, and expression of these pro-inflammatory cytokines is often correlated with a longer-lasting inflammatory

**Fig. 4 Single-cell RNA-seq analysis of hiPSC microglial chimeric mouse brain. a** A schematic diagram showing the experimental design. Microglia were isolated from the highlighted brain regions at 6 months post transplantation. **b** tSNE plot of 11 cell types as identified by characteristic cell-specific gene expression, following translation of human gene symbols to mouse symbols as described in "Methods". Arrow indicates the human xenograft microglia (Xeno MG). **c** Dot plot showing two representative cell-specific genes for each cell cluster. As indicated by the legend, the diameter of the dot indicates the percent of cells within a cluster expressing the gene (percent expression). The color saturation indicates the average expression over the cluster (average expression; log normalized counts). The cluster numbers, colors of clusters in **b**, and selected cell identities are shown at left. **d** tSNE plots with dots (representing individual barcodes/cells) colored by expression of canonical microglial genes, based on expression level determined in Seurat (log normalized counts). **e** A heatmap showing expression of signature genes of hematopoietic progenitor cells (HPC) and microglia in PMPs and Xeno MG, showing individual replicates ($n = 3$ for PMP, $n = 4$ averaged human clusters for Xeno MG). Color indicates the expression level normalized for each gene using a Z-score. **f** PCA plot of our Xeno MG, hiPSC-derived PMPs, and individual cell RNA-seq expression data from publicly available datasets, including hiPSC-derived microglia cultured under 2-dimensional (2D) conditions (iPS MG)[12, 54], hiPSC-derived microglia developed in 3D cerebral organoids (oMG)[55], hiPSC-derived microglia developed for 2–7.8 months in mouse brain (xMG)[65]. Brain tissue-derived adult human microglia, including adult ex vivo microglia (Adult MG ex vivo) from Gosselin et al.[7] (age from 13 to 17 years) and Galatro et al. (age from 34 to 102 years)[6], in vitro microglia (MG In Vitro) from Gosselin et al.[7], as well as blood/liver macrophages[12, 54]. **g** tSNE plots of selected human cells showing the expression of *CD74, SPP1, C3, BAZ2B, CCL4, EGR2, EGR3,* and *CD83* gene transcripts. *CD74, SPP1, C3,* and *BAZ2B* appear to be uniformly expressed in all cells, but *CCL4, EGR2, EGR3,* and *CD83* are enriched in distinct subsets of in Xeno MG.

response[53]. This observation suggests that only a very mild inflammatory reaction was likely triggered in the Xeno MG during sample preparation. These results demonstrate that Xeno MG developed in the mouse brain largely retain their human microglial identity and exhibit a gene expression pattern characteristic of homeostatic human microglia.

Next, to further evaluate identity and maturation of Xeno MG in chimeric mouse brains, we compared the global expression patterns of 21 genes including the 11 microglia-specific genes and seven HPC-specific genes shown in Fig. 4e, as well as three NPC genes (*NES, DCX,* and *SOX2*) between our Xeno MG, hiPSC-derived PMPs and published datasets of hiPSC-derived microglia cultured under 2-dimensional (2D) conditions (iPS MG)[12,54], hiPSC-derived microglia developed in 3D cerebral organoids (oMG)[55], hiPSC-derived microglia developed in mouse brain (xMG) as reported in a recent study[65], brain tissue-derived adult human microglia, including adult ex vivo microglia (Adult MG ex vivo) from Gosselin et al.[7] (age from 13 to 17 years) and Galatro et al. (age from 34 to 102 years)[6], in vitro microglia (MG in vitro) from Gosselin et al.[7], as well as blood/liver macrophages[12,54]. As shown in Fig. 4f, a PCA demonstrated that Xeno MG, together with xMG, were markedly distinct from blood/liver macrophages, PMPs, and the hiPSC-derived MG cultured under 2D conditions or developed in organoids. The human microglia cultured in vitro were separate from the other two clusters, which might suggest the significant impact of culture conditions on gene expression in those microglia as previously reported[7]. Our Xeno MG clusters intermingled with a cluster of adult MG, suggesting their resemblance to adult human microglia. Recent unbiased hierarchical clustering analyses revealed four major subclasses of adult human microglia derived from human brain tissue[56]. To determine if Xeno MG also exhibited similar heterogeneity in chimeric mouse brain, we examined the expression of the most differentially regulated genes identified from the different subclasses of adult human microglia[56]. Gene expression analysis revealed that *CD74, SPP1, C3,* and *CST3*, which were highly expressed in all subclasses in adult human microglia, had a similarly uniform pattern of expression among most Xeno MG cells. Moreover, a chemokine gene *CCL4*, the zinc finger transcription factors *EGR1, EGR2* and *EGR3*, *CD83*, and *MCL1*, which are each characteristically expressed in individual subclasses of human microglia, similarly had upregulated expression in distinct subpopulations of Xeno MG (Fig. 4g and Supplementary Fig. 8C). Taken together, these results demonstrate that Xeno MG developed in the mouse brain highly resemble mature human microglia and faithfully recapitulate heterogeneity of adult human microglia.

**Differences between coresident Xeno MG and mouse microglia.** Previous studies reported differences in transcriptomic profiles between human and mouse microglia[6,7]. In the chimeric mouse brain, as xenografted hiPSC-derived microglia and host mouse microglia developed in the same brain environment, this model may provide a unique opportunity to directly examine the differences between human and mouse microglia. Xeno MG and host mouse microglia clusters obtained from four independent samples of 6-month-old chimeric mouse brains were used for the following comparison (Fig. 5). We first compared the average levels of microglial gene transcripts in Xeno MG with orthologous gene transcripts in host mouse microglia. Consistent with previous findings[6,7], the comparison between Xeno MG and mouse microglial transcriptomes demonstrated similar gene expression patterns overall ($r^2 = 0.553$; $p < 2.2 \times 10^{-16}$), and the majority of orthologous genes pairs (14,488 of 15,058; 96.2%) were expressed within a twofold range (black dots, Fig. 5b). Using a cut-off of twofold difference and an FDR of 0.05, we identified that 91 gene transcripts were preferentially expressed in human microglia, whereas 84 gene transcripts were preferentially expressed in mouse microglia (Supplementary Fig. 8D and Supplementary Data 2). Importantly, previously reported signature genes expressed in human microglia[7], including *SPPI, A2M,* and *C3*, and signature genes expressed in mouse microglia, including *HEXB, SPARC,* and *SERINC3*, were all differentially expressed in our sequencing data (Fig. 5b, c), indicating the high fidelity of our samples in resembling previously identified human versus mouse microglial gene expression profiles. To explore the function of genes that were highly expressed human microglia, we further performed gene ontology (GO) term analysis. Many significantly enriched terms were associated with the innate immune activity of microglia, such as "immune system response," "cellular response to chemical stimulus," and "regulation of cytokine production." This finding of enriched innate immunity-related gene in our Xeno MG could be either reflect the nature of human microglia as reported previously[6,57], or the result from differential responses to critical signals from murine molecules, such as fractalkine. These results suggest that, compared with the host mouse microglia, Xeno MG and mouse microglia exhibit overall similar patterns of transcriptomic profile, but numerous species-specific differentially expressed genes were also observed.

Previous studies have shown that several disease-risk genes, such as genes associated with AD, Parkinson's disease (PD), multiple sclerosis (MS), and schizophrenia, are preferentially expressed in microglia[7,57]. Moreover, relative expression of these genes in human and mouse microglia are also different[8]. Therefore, we examined

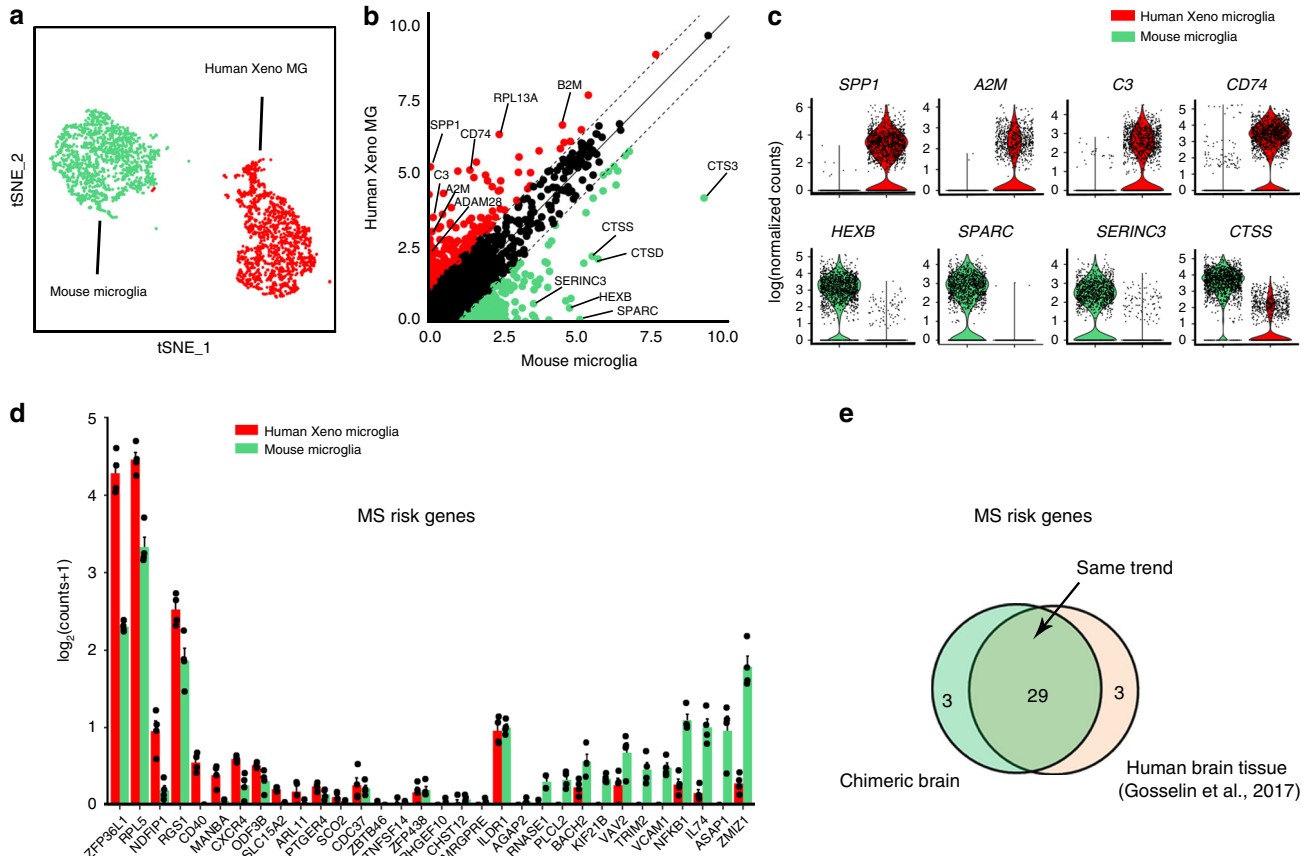

**Fig. 5 Transcriptomic profiling analysis of clusters of Xeno MG and mouse microglia. a** tSNE plot highlighting only the clusters of human Xeno MG and mouse host microglia. **b** Scatter plot showing mean mRNA expression levels of human and mouse genes with unique orthologs from Xeno MG and mouse microglia clusters, highlighting the differentially expressed genes (DEGs; at least twofold different) in human Xeno MG (red) or mouse microglia (green) from 6-month-old chimeric mouse brain. Significantly different DEGs (<5% false discovery rate (FDR) and at least twofold different) are listed in Supplementary Data 2. **c** Violin plots summarizing expression differences in individual cells within the human Xeno MG and mouse microglia clusters. Dots indicate expression levels (as log normalized counts) of individual cells and the violin shape summarizes the distribution of expression in the population. **d** Bar plots showing the average expression (mean ± SEM, $n = 4$ samples) of Multiple sclerosis (MS)-associated genes in Xeno MG and mouse microglia. These genes were reported to be differentially expressed between human and mouse microglia as in Gosselin et al.[7]. Source data are provided as a Source Data file. **e** Venn diagrams showing that majority of the MS associated genes that were reported to be differentially expressed between human and mouse microglia are recapitulated in our chimeric mouse model (29 of 32).

the expression of disease-risk genes in Xeno MG and mouse microglia from our chimeric mouse brain preparation. Expression of disease-risk genes, as reported in a recent study[7], had a highly similar differential expression pattern in coresident mouse and human microglia (Fig. 5d, e and Supplementary Fig. 8F–I). Specifically, with respect to MS, we found that out of 32 MS genes, 29 genes, including *ZFP36L1*, *RPL5*, and *NDFIP1* were more abundantly expressed in Xeno MG than in mouse microglia (Fig. 5d, e). Similarly, out of 14 AD genes, ten genes, including *Apoc1, Sorl2*, and *Mpzl1*, were more abundantly expressed in Xeno MG than in mouse microglia (Supplementary Fig. 8F, I). Out of the 20 PD genes listed in a previous report[7], 18 genes, such as *Vps13c, Snca, Fgf20, Mnnrn1*, and *Lrrk2*, had the same trend of differential expression with greater expression in Xeno MG than in mouse microglia (Supplementary Fig. 8H, I). We also found that some of the disease genes were preferentially expressed in mouse microglia, such as *Syt11* and *Gba* in PD. Altogether, these observations demonstrate that our hPSC microglial chimeric mouse brain can faithfully model disease-relevant transcriptomic differences between human and mouse microglia, and this model will serve as a unique tool for modeling human neurological disorders that involve dysfunction of microglia.

**Responses of Xeno MG to cuprizone-induced demyelination.** To explore whether Xeno MG are functionally dynamic in response to insult, we fed 3-month-old chimeric mice with cuprizone-containing diet to induce demyelination. The cuprizone model is one of the most frequently used models to study the pathophysiology of myelin loss in MS[58]. It is appropriate to use our hiPSC microglial chimeric mouse brain to examine the dynamics of human microglia under a demyelination condition, considering our observation that a large number of Xeno MG reside in the corpus callosum at 3–4 months post transplantation and Xeno MG were found nearly exclusively in the corpus callosum at 6 months post transplantation (Fig. 2c). After 4 weeks of cuprizone treatment, we found that myelin structure, indicated by MBP staining in the corpus callosum, was disrupted and became fragmented in our chimeric mice (Fig. 6a), in contrast to the intact and continuous MBP⁺ myelin structure in chimeric mice fed with control diet (Supplementary Fig. 9). As shown in the super-resolution images in Fig. 6b, c, engulfment of MBP⁺ myelin debris by both Xeno MG and mouse microglia were clearly seen in the corpus callosum. Notably, more myelin debris was found inside of mouse microglia, compared with Xeno MG (Fig. 6d). In addition, we also examined the expression of CD74

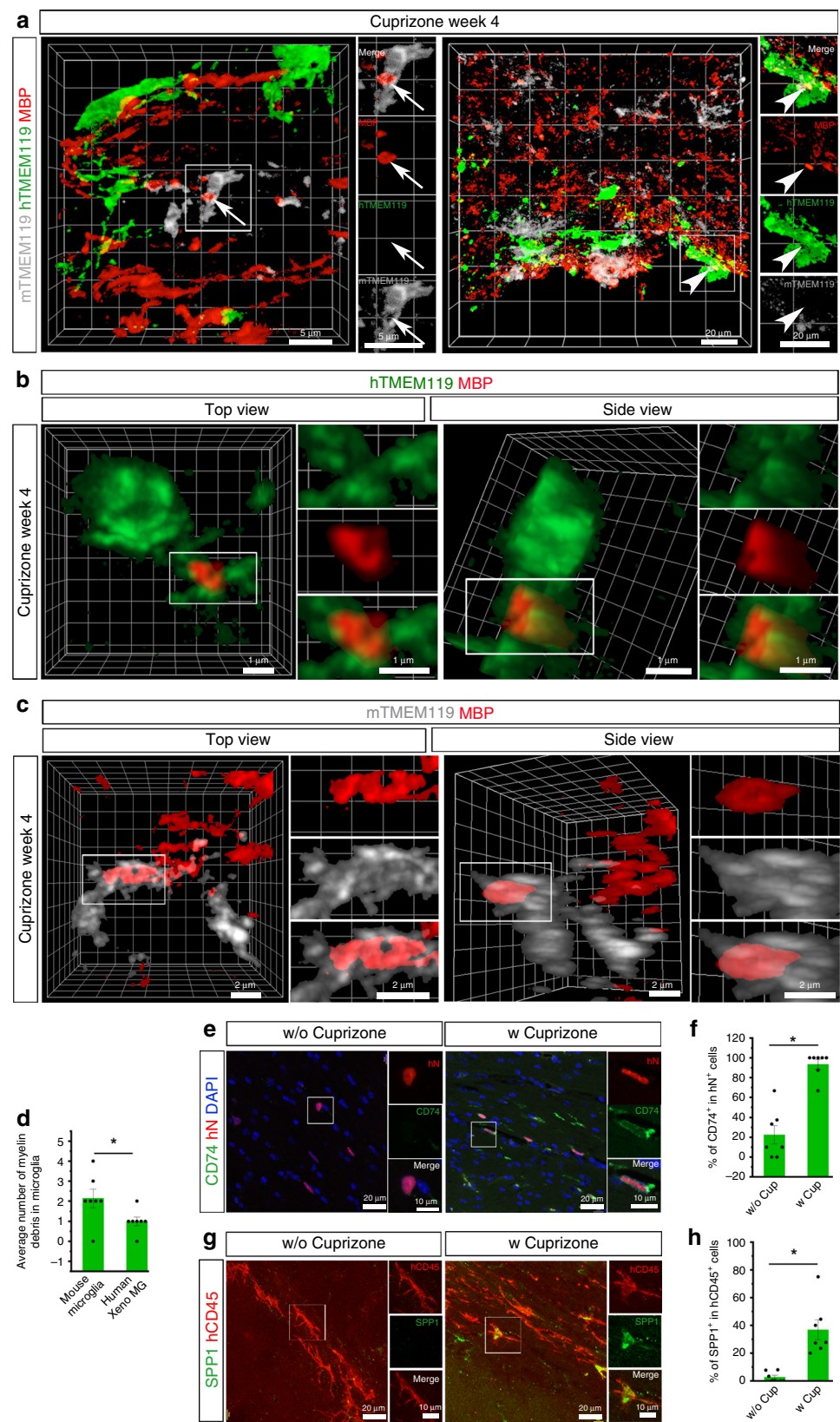

and SPP1, which is known to be upregulated in MS[56]. Without cuprizone treatment, variations in CD74 expression among animals were observed in Xeno MG and on average, about 20% of Xeno MG expressed CD74 (Fig. 6e, F). Nearly no Xeno MG

expressed SPP1 in the corpus callosum (Fig. 6g, h). With cuprizone treatment, many of the Xeno MG expressed CD74 or SPP1, recapitulating the upregulated expression of CD74 and SPP1 in MS (Fig. 6e–h). In mice, CD74 and SPP1 were shown to

**Fig. 6 Responses of hPSC-derived microglia to cuprizone-induced demyelination. a** Representative 3D reconstruction images showing hTMEM119+ donor-derived microglia and mTMEM119+ mouse host microglia interact with MBP+ myelin debris in the corpus callosum after 4 weeks of cuprizone diet. Scale bars, 5 or 20 μm in the original or enlarged images as indicated. Representative 3D reconstruction of super-resolution images showing hTMEM119+ donor-derived microglia (**b**) and mTMEM119+ mouse host microglia (**c**) engulf MBP+ myelin debris in the corpus callosum after 4 weeks of cuprizone diet. Scale bars, 1 μm in the original or enlarged images. **d** Quantification of average number of myelin debris in mouse host microglia and human Xeno MG ($n = $ 7 mice). The data are pooled from the mice that received transplantation of microglia derived from both hESCs and hiPSCs. Student's $t$ test (two-sided). *$P < 0.05$. Data are presented as mean ± s.e.m. **e** Representative images showing CD74+ and hN+ cells after 4 weeks of diet with (w) or without (w/o) cuprizone. Scale bars, 20 and 10 μm in the original or enlarged images, respectively. **f** Quantification of the percentage of CD74+ cells in total hN+ cells after 4 weeks of diet with (w) or without (w/o) cuprizone ($n = 7$ mice). The data are pooled from the mice that received transplantation of microglia derived from both hESCs and hiPSCs. Student's $t$ test (two-sided). *$P < 0.05$. Data are presented as mean ± s.e.m. **g** Representative images showing SPP1+ and hN+ cells after 4 weeks of diet with (w) or without (w/o) cuprizone. Scale bars, 20 and 10 μm in the original or enlarged images, respectively. **h** Quantification of the percentage of SPP1+ cells in total hN+ cells after 4 weeks of diet with (w) or without (w/o) cuprizone ($n = 7$ mice). The data are pooled from the mice that received transplantation of microglia derived from both hESCs and hiPSCs. Student's $t$ test (two-sided). *$P < 0.05$. Data are presented as mean ± s.e.m. Source data are provided as a Source Data file.

characterize white matter-associated subsets of microglia that appear during development[33]. In our study, there was a discrepancy between the *CD74* and *SPP1* transcript levels and their protein levels. This might suggest an involvement of a sophisticated control of mRNA translation that was observed in the regulation of other genes critical for brain development[59,60]. Altogether, these results demonstrate that human Xeno MG are dynamic in response to insult in mouse brain.

## Discussion

In this study, by engrafting neonatal mice with hPSC-derived PMPs, we demonstrate the generation of chimeric mouse brains in which hPSC-derived microglia widely disperse. We propose that the following three reasons may account for the generation of human microglial chimeric mouse brain. First, as compared with other types of neural cells, microglial cells are unique in that they turn over remarkably quickly, allowing the vast majority of the population to be renewed several times during a lifetime[61–63]. Previous studies have shown that neonatally transplanted human macroglial or neural progenitor cells can outcompete and largely replace the host mouse brain cells[17,18,64]. In this study, we also observe that the hPSC-derived PMPs are highly proliferative prior to transplantation and transplanted cells divide for at least 6 months in the mouse host brain. Therefore, the nature of high turnover rate of microglia and the competitive advantage of engrafted human cells over endogenous mouse cells may result in a large number of human donor-derived microglia and brain regions being repopulated by hPSC-derived microglia in the mouse brain at 6 months. Second, during early brain development, microglial cells use blood vessels and white matter tracts as guiding structures for migration and enter all brain regions[24]. Thus, transplantation of hPSC-derived PMPs to the anterior anlagen of the corpus callosum of the neonatal mouse brain in this study may facilitate migration of donor cell-derived microglia, resulting in wide dispersion of hPSC-derived microglia into different brain regions. In addition, in support of this concept, we also observe that in the mouse brain, hPSC-derived microglia are concentrated around and have close contact with blood vessels in both gray matter and white matter. Lastly, we transplanted hPSC-derived PMPs into the mouse brain at the earliest postnatal age, P0, as in general the neonatal brain is more receptive for the transplanted cells and more conducive for their survival and growth[15,17]. This is also supported by a recent study in which neonatal animals were used for cell transplantation[65]. Moreover, we characterized the donor-derived microglia up to 6 months post transplant, which allowed the donor cells to develop for a long term in the mouse brain. Although there were variations in chimerization among animals, this human microglia chimera

model is highly reproducible according to scRNA-seq analysis using four chimeric mouse brains. We calculated the numbers of detected mouse/human microglia in each mouse brain sample and found that human microglia were consistently detected in each sample (legend to Supplementary Fig. 7). In addition, the high reproducibility of generating such a hiPSC microglial chimeric mouse brain model was also corroborated by two other recent reports[65,66].

Remarkably, we find that xenografted hPSC-derived microglia developed in the mouse brain retain a human microglial identity. Importantly, xenografted hPSC-derived microglia showed expression patterns of microglial maturity resembling adult human microglia derived from human brain tissue. Therefore, establishment of such a hPSC microglial chimeric mouse model provides unique opportunities for understanding the biology of human microglia. First, this proof-of-concept study paves the path to interrogating the species-specific differences between human versus mouse microglia at molecular, functional, and behavioral levels using this hPSC microglia chimeric mouse brain model. Microglia are intimately involved in processes of neuronal development, such as neurogenesis, synaptogenesis, and synaptic pruning[1]. Building upon the differential expression profiles, our model will be useful to investigate how human and mouse microglia function differently in shaping neuronal development. Similar to a recent report[65], our engrafted PMPs give rise to a small population of CNS macrophages. There are also potentially mouse-derived CNS macrophages in the brain tissue that we collected for single-cell RNA-seq, but likely to represent a small fraction of the total cluster. Therefore, within microglial clusters from each species, results may include some similar but distinguishable cell types. However, the primary gene expression differences are likely to be driven by the majority microglia. This was also supported by the observation that previously reported signature genes expressed in human versus mouse microglia[7] were differentially expressed in our results. Nevertheless, we take a conservative interpretation and did not claim to identify any novel gene expression differences between human versus mouse microglia based on the current RNA-seq data. Second, a previous study reports that engrafted human astrocytes modulate other CNS cell types in the mouse brain, particularly enhancing neuronal synaptic plasticity[17]. Thus, in future studies, our hPSC microglial chimeric mouse model will provide fascinating opportunities to understand how the inclusion of human microglia in the developing brain ultimately impacts neuronal development, synaptic plasticity, as well as the behavioral performance of the animals. Third, several transcriptomic studies[6,14] have clearly demonstrated that microglial genes are differently regulated during aging and neurodegeneration between mice and

humans, indicating the importance of developing a human microglia model to study human microglial function across different development stages, particularly adult microglia for studying aging-related and neurodegenerative disorders. Our scRNA-seq data suggests that Xeno MG resemble adult human microglia. Moreover, functional analyses demonstrate that Xeno MG are dynamic in response to cuprizone-induced demyelination and exhibit the upregulated expression of CD74 and SPP1 seen in MS patients[56], providing proof of concept that our chimeric mouse model has the ability to recapitulate functional changes of human microglia under disease conditions.

Similar to reports of macroglial or neuronal chimeric mouse brain models[15–17], in the current hPSC microglial chimeric mouse model, the endogenous mouse counterpart cells are still present. In contrast to macroglial cells and neurons, microglial cells can be acutely depleted (up to 99% depletion) in the entire brain without significantly affecting the viability of animals, by pharmacologically inhibiting signaling pathways that are important for the survival and development of microglia, such as CSF1 signaling[67] or by genetically coupling suicide genes under the control of promoters of microglia-specific genes[1,68]. In future studies, it will be interesting to explore the possibility of creating "humanized" mouse brains containing solely hPSC-derived microglia, by depleting endogenous mouse microglia using pharmacological or genetic approaches in neonatal mouse brains prior to engraftment of hPSC-derived microglia. Combined with recently developed hPSC cerebral organoid models that contain microglia[11,55], chimeric mouse brain models may help further our understanding of the complex interactions between human microglia and human neurons and macroglial cells under normal and disease conditions.

## Methods

**Generation, culture, and quality control of hPSC lines.** One healthy control female hiPSC line and female H9 ESC line were used this study. The hiPSC line were generated from healthy adult-derived fibroblasts using the "Yamanaka" reprogramming factors[69]. The hPSC line has been fully characterized by using karyotyping, teratoma assay, DNA fingerprinting STR (short tandem repeat) analysis, gene expression profiling, and Pluritest (www.PluriTest.org)[16,69]. The hPSCs were cultured on dishes coated with hESC-qualified Matrigel (Corning) in mTeSR1 media (STEM CELL Technologies) under a feeder-free condition. Human PSCs from passage number 15–30 were used. The hPSCs were passaged with ReLeSR media (STEM CELL Technologies). All hPSC studies were approved by the Embryonic Stem Cell Research Oversight committee at Rutgers University.

**PMP generation and culture.** PMP were generated from hiPSCs and H9 hESCs, using a published protocol[13]. Briefly, the yolk sac embryoid bodies (YS-EBs) were generated by treating the EBs with bone morphogenetic protein 4 (BMP4, 50 ng/ml, Peprotech; to induce mesoderm), vascular endothelial growth factor (50 ng/ml, Peprotech; endothelial precursors), and stem cell factor (SCF, 20 ng/ml, Miltenyi Biotech; hematopoietic precursors). Next, the YS-EBs were plated into dishes with interleukin-3 (IL-3, 25 ng/ml, Peprotech) and macrophage colony-stimulating factor (100 ng/ml, Invitrogen) to promote myeloid differentiation. At 2–3 weeks after plating, human hPMPs emerged into the supernatant and were continuously produced for more than 3 months. The cumulative yield of PMPs was around 40-fold higher than the number of input hPSCs (Fig. 1a), similar to the previous studies[11,13]. PMPs were produced in a Myb-independent manner and closely recapitulating primitive hematopoiesis[1,13,19].

**Animals and cell transplantation.** PMP were collected from supernatant and suspended as single cells at a final concentration of 100,000 cells per μl in PBS. The cells were then injected into the brains of P0 Rag2$^{-/-}$hCSF1 immunodeficient mice (C;129S4-Rag2$^{tm1.1Flv}$ Csf1tm1$^{(CSF1)Flv}$ Il2rg$^{tm1.1Flv}$/J, The Jackson Laboratory). The precise transplantation sites were bilateral with the midline = ±1.0 mm, posterior bregma = −2.0 mm, and dorsoventral depth = −1.5 and −1.2 mm (Fig. 1c). The mouse pups were placed on ice for 5 min to anesthetize. The pups were injected with 0.5 μl of cells into each site (total four sites), using a digital stereotaxic device (David KOPF Instruments) that was equipped with a neonatal mouse adapter (Stoelting)[16]. The pups were weaned at 3 weeks and were kept up to 6 months. All animal work was performed without gender bias with approval of the Rutgers University Institutional Animal Care and Use Committee.

Both hiPSC- and hESC-derived PMPs were transplanted into mouse brains. Both engrafted hiPSC- and hESC-derived microglia were analyzed, including characterization of their marker expression (Fig. 1), morphological changes along brain development (Fig. 2), and their phagocytic functions under homeostatic condition (Fig. 3), as well as toxin-induced demyelination condition (Fig. 6). For the quantification, we pooled the data collected from both hiPSC- and hESC-derived microglia. In the single-cell RNA-sequencing experiment, we only used the animals received transplantation of hiPSC-derived microglia. For cuprizone-induced demyelination, 3-month-old chimeric mice were fed with cuprizone diet (Sigma-Aldrich, 0.2%) or control diet for 4 weeks.

**Sample preparation and library construction for scRNA-seq.** Six months old chimeric mice that received transplantation of microglia derived from the hiPSC line were used for single-cell RNA-sequencing experiments. The mice were perfused with oxygenated solution (2.5 mM KCl, 87 mM NaCl, 1.25 mM NaH$_2$PO$_4$, 26 mM NaHCO$_3$, 75 mM sucrose, 20 mM glucose, 2 mM MgSO$_4$, and 1 mM CaCl$_2$) and the brain was quickly extracted and kept in the same cold solution for vibratome (VT1200, Leica) sectioning (500 μm thickness) and dissection. The brain regions were isolated from where engrafted hiPSCs-derived microglia largely dispersed, including the cerebral cortex, hippocampus, corpus callosum, and olfactory bulb. The selected regions were chopped with Spring Scissors (WPI) into fine pieces for further dissociation into single cells, based on 10× Genomic Sample Preparation Domstratd Protocol (Dissociation of Mouse Embryonic Neural Tissue) with modifications. Briefly, the pieces were collected and dissociated with the Papain (1 mg/ml, Sigma) and DNAase I (100 unit/ml, Roche) in Hibernate solution (Gibco) in 37 °C for 20 min. Tissues were washed and triturated with wide bore tips in cold Hibernate solution until no visible chunks. The samples were spun down in 200 rcf for 2 min in 4 °C and filtered through 30 μm cell strainer to obtain single cells for cell counting and library preparation. To generate libraries, 20,000 cells were loaded per sample. Chromium™ Single Cell 3′ Library and Gel Bead Kit v2, 4 rxns, Chromium™ i7 Multiplex Kit, 96 rxns, and Chromium™ Single Cell A Chip Kit, 16 rxns are used from 10× Genomic single-cell gene expression kit. cDNA libraries were generated following the manufacturer's instructions.

**Bulk and scRNA-seq.** We performed bulk RNA-sequencing analysis of hiPSC-derived PMPs[16]. Total RNA was prepared using RNAeasy kit (QIAGEN). We used 600 ng of total RNA from each sample to construct libraries. The libraries were constructed using the TruSeqV2 kit from Illumina (Illumina, San Diego, CA). The libraries were subjected to 75 bp paired read sequencing using a NextSeq500 Illumina sequencer. Approximately, 30–36 million paired reads were generated for each sample. Bc12Fastq software, version 1.8.4 was used to generate Fastq files. The genome sequence was then indexed using the rsem-prepare-reference command. Each fastq file was trimmed and checked for quality with fastp (v. 0.12.2)[70], and then aligned to the UCSC hg38 human genome using Hisat2 (v.2.1.0)[71,72]. Transcript counts were extracted using the featureCounts function of the Rsubread package[73].

For scRNA-seq, we used Chromium™ i7 Multiplex Kit, 96 rxns, Chromium™ Single Cell 3′ Library and Gel Bead Kit v2 and Chromium™ Single Cell A Chip Kit for capture and library preparation. scRNA-seq was performed by RUCDR® Infinite Biologics at Rutgers by using a 10× Genomics single-cell gene expression profiling kit. The libraries were analyzed on Agilent 4200 TapeStation System using High Sensitivity D1000 ScreenTape Assay (Cat #: 5067-5584) and quantified using KAPA qPCR (Cat #: KK4835). Libraries and then normalized to 10 nM before being pooled together. The pooled library was then clustered and sequenced on Illumina HiSeq 2500 in Rapid Run Mode, using the following parameters: 36 bp forward read, 100 bp reverse read, and 8 bp index read. For each individual library, the sequencing data from four unique indexes were combined before further analysis.

Sequencing reads were aligned with pooled mouse (mm10) and human (hg19) reference genomes and the barcodes were interpreted using Cellranger software (10× Genomics, v. 3.0.0). The resulting matrices of gene counts × barcodes were coded by individual sample identifier and loaded into Seurat (v. 3.0.0) software[74–76] in R/Bioconductor[77]. An initial analysis revealed a distinct cluster of human-expressing cells. To compare expression across species, a strategy was employed similar to one used previously[78]. A table of 17,629 unique matching genes was prepared, starting with a human-mouse gene homology list obtained from Jackson Labs (http://www.informatics.jax.org/downloads/reports/index.html#marker), and hand-curating to remove duplicates. The homologous genes list is deposited in a public data archive at https://github.com/rhart604/mousify/blob/master/geneTrans.txt. Sample/barcode identifiers for the human-specific data were isolated and matching gene symbols were converted from human to mouse. Sample/barcode identifiers not matching this cluster were assumed to be mouse, and these were trimmed to retain only mouse gene symbols matching the homology list. Complete details of homology gene translation are described elsewhere[79].

To model both human and mouse results in a comparable system, we assigned individual sequencing reads to the optimal species using the alignment score in the bam file, and then separated the original fastq files into individual sets specific by species. These were re-processed with separated mouse or human reference genome indices and then recombined after translating homologous genes. In the end, only 147 out of 19,154 barcodes, or 0.76%, included sequences optimally

aligning with both species, likely caused by the creation of droplets containing cells from more than one species, so these were eliminated from further consideration. The entire procedure, along with comparisons with alternative strategies, including all required R and Python code, is described elsewhere[79].

For analysis of human microglial sub-clusters, extracted human sample/barcode were restricted to human gene symbol results and re-analyzed with Seurat. GO analysis used the g:Profiler[80] website (https://biit.cs.ut.ee/gprofiler/gost).

For comparisons among sources of human microglia, raw RNA-seq reads from the human-specific cluster were pooled by sample and aligned with reference human genome (hg38) using HISAT2 (ref. [72]). Raw sequencing reads from other publications were downloaded from GEO (series accessions GSE99074 (ref. [6]), GSE97744 (ref. [12]), GSE102335 (ref. [55]), and GSE133434 (ref. [65])). After similar HISAT2 alignment, all count summaries were imported into a DESeq2 (ref. [81]) data model and a variance stabilization transformation was applied prior to principal components analysis.

**Immunostaining and cell counting**. Mouse brains fixed with 4% paraformaldehyde were put in 20 and 30% sucrose for dehydration. After dehydration, brain tissues were blocked with OCT and frozen by solution of dry ice and pure alcohol. The frozen tissues were cryo-sectioned with 30-μm thickness for immunofluorescence staining[82]. The tissues were blocked with blocking solution (5% goat or donkey serum in PBS with Triton X-100) in room temperature (RT) for 1 h. The Triton X-100 concentration was 0.8% for brain tissue. The primary antibodies were diluted in the same blocking solution and incubated with tissues in 4 °C overnight. The primary antibodies were listed in Supplementary Table 1. Then, the sections were washed with PBS and incubated with secondary antibodies for 1 h in RT. After washing with PBS, the slides were mounted with the anti-fade Fluoromount-G medium containing 1, 40,6-diami-dino-2-phenylindole dihydrochloride (DAPI) (Southern Biotechnology). For PSD95 peptide blocking experiment, PSD95 antibody (2.5 μg/ml) was incubated with five times PSD95 blocking peptide (12.5 μg/ml, Synaptic System) or equivalent amount of PBS at RT for 30 min before applying secondary antibodies to the brain tissues as suggested by the manufacturer instructions.

Images were captured with a Zeiss 710 confocal microscope. Large scale images in Fig. 1D, Supplementary Fig. 1A, B were obtained by confocal tile scan and automatically stitched using 10% overlap between tiles by Zen software (Zeiss). For 3D reconstruction images in Figs. 3e and 6a and Supplementary Fig. 6B, D, and E were processed by Zen software (Zeiss). To visualize synaptic puncta pruning, myelin debris engulfment, super-resolution images in Figs. 3a, b, d and 6b and 4c, Supplementary Figs. 5 and 6A, C were acquired by Zeiss Airyscan super-resolution microscope at 63X with 0.2 mm z-steps. To generate 3D-surface rendered images, super-resolution images were processed by Imaris software (Bitplane 9.5). To visualize the engulfed PSD95$^+$ puncta within microglia, any fluorescence that was outside of the microglia was subtracted from the image by using the mask function in Imaris[83]. Cell number and microglia process length and endpoints were counted with ImageJ software[16]. At least five consecutive sections of each brain region were chosen. The number of positive cells from each section was counted after a Z projection and at least seven mice in each group were counted. The number of synaptic puncta or myelin debris were calculated from seven mice in each group, minimal ten cells per animal. Engraftment efficiency and degree of chimerization were assessed by quantifying the percentage of hTMEM119$^+$ cells among total DAPI$^+$ cells in sagittal brain sections, as reported in the previous studies[17,18]. The cell counting was performed on every fifteenth sagittal brain section with a distance of 300 μm, covering brain regions from 0.3 to 2.4 mm lateral to the midline (seven to eight sections from each mouse brain were used).

**Statistics and reproducibility**. All data represent mean ± s.e.m. When only two independent groups were compared, significance was determined by two-tailed unpaired $t$ test with Welch's correction. When three or more groups were compared, one-way ANOVA with Bonferroni post hoc test or two-way ANOVA was used. A $P$ value <0.05 was considered significant. The analyses were done in GraphPad Prism v.5. All experiments were independently performed at least three times with similar results.

**Reporting summary**. Further information on research design is available in the Nature Research Reporting Summary linked to this article.

## Data availability
Source data underlying Figs. 1, 2, 3, 5, and 6 and Supplementary Figs. 2 and 8 are available as a Source Data file. All other relevant data supporting this study are available from the corresponding author upon reasonable request. The GEO accession numbers of the single-cell RNA sequencing data and bulk sequencing data reported in this study are: GSE129178 and GSE139161.

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

## Acknowledgements

This work was in part supported by grants from the NIH (R21HD091512 and R01NS102382 to P.J.). R.P.H. was supported by R01ES026057, R01AA023797, and U10AA008401. X.L. was supported by the Jiangsu Government Scholarship for Overseas Studies from China. A.J.B is supported by the National Institute of General Medicine Sciences (NIGMS) NIH T32 GM008339. We thank Dr Cheryl Dreyfus from Rutgers Robert Wood Johnson Medical School for the suggestions on cuprizone model, Dr Sally A. Cowley from Oxford Stem Cell Institute for the suggestions on inducing microglia differentiation, Dr Noriko Kane-Goldsmith from Rutgers University for assistance with Zeiss Airyscan super-resolution imaging, and Dr Brian Daniels from Rutgers University for critical reading of the manuscript. We also thank Ms Maharaib Syed from Rutgers University for the assistance with immunohistochemistry.

## Author contributions

P.J. and R.X. designed experiments and interpreted data. R.X. carried out most of experiments with technical assistance from X.L., A.J.B., A.P., and K.K.. R.P.H. performed the gene expression analysis, interpreted the data, and provided critical suggestions to the overall research direction. P.J. directed the project and wrote the manuscript together with R.X. and input from all co-authors.

## Competing Interests

The authors declare no competing interests.
