## [Peer Review File · Nature Communications]

Point-by-point Response to Reviewers' Comments:

We thank all the reviewers for their critical and constructive suggestions. While our manuscript was under revision, a paper was published in Neuron (Hasselmann et al., Neuron. 2019, PMID: 31375314), reporting the generation of a similar chimeric mouse brain model and single-cell RNA-seq analysis of the chimeric mouse brain. Although our study and the Hasselmann et al. paper corroborate each other, our revised manuscript distinguishes ours from the previous report, with an added emphasis on functional characterization of the engrafted human pluripotent stem cell (hPSC)-derived microglia not only under homeostatic conditions, but also under a toxin-induced demyelination condition. In the revised manuscript, complemented by the newly generated RNA-seq data from hPSC-derived primitive macrophage progenitors, we optimized analysis of the single-cell RNA-seq data to strongly support the microglial identity of our engrafted cells. Importantly, we have performed new experiments, particularly employing super-resolution confocal imaging techniques, to examine phagocytic functions of engrafted human microglia under homeostatic and toxin-induced demyelination conditions.

In our revised manuscript, we have shifted the focus to functional characterization, which echoes with Reviewer #2's comment that *"The finding that human macrophages can engulf mouse synapses, however, is a major finding. It would be the first validation that human microglia-like cells perform microglia-specific functions in vivo"*. Therefore, we also changed the title of our study to "Functional Mature Human Microglia Developed in Human iPSC Microglial Chimeric Mouse Brain".

We feel that these additional data have indeed strengthened our paper and for that we greatly appreciate the reviewers' suggestions. We have revised our manuscript and made modifications throughout the text but, for clarity, we highlighted the major changes in red in the revised manuscript.

Reviewer #1:

Remarks to the Author:

Microglia play critical roles in normal brain function and are implicated in disease states of the central nervous system. Due to the difficulty of obtaining human microglia samples, rodent models are often used. However, data show rodent microglia do not perfectly mimic their human counterparts, leaving a gap in the translatability of rodent microglial studies. Xu et al. describe a novel chimeric mouse model where they transplant human pluripotent stem cell (hPSC)-derived macrophage precursors into the brains of neonatal immunocompromised mice. The human cells disperse throughout the mouse brain and replace murine microglia in the corpus callosum, cortex, olfactory bulb, and cerebellum up to 6 months post-transplant. These hPSC-derived microglia are more representative of normal human microglia by gene expression patterns as well as by analyzing transcriptomes of CNS disease-risk genes.

The authors present a promising model showing that engraftment of human macrophage precursors in neonatal mice better represent normal human microglia. Although the engraftment of human microglia in this manuscript seems greater than that of other published reports, the overall results are incremental to previous reports engrafting iPSC derived human microglia into similar mice (Abud et al 2017; McQuade 2018) and reports comparing the transcriptomes of human versus mouse microglia.

Below are specific concerns that should be addressed.

1. It is stated that hPSC-induced microglia were derived either from hiPSCs or a human Embryonic Stem Cell line. It's not clear in the results which of these cells were injected then further analyzed after engraftment.

⇒ **Response:** To clarify, we derived microglia from one hiPSC line and one human embryonic stem cell (hESC) line in this study. Both hiPSC- and hESC-derived microglia were transplanted into mouse brains. Both engrafted hiPSC- and hESC-derived microglia were analyzed, including characterization of their marker expression (**Figure 1**), morphological changes along brain development (**Figure 2**), as well as their phagocytic functions under homeostatic (**new Figures 3**) and toxin-induced demyelination conditions (**new Figure 6**). For the quantification, we pooled the data collected from both hiPSC- and hESC-derived microglia. In the single-cell RNA-sequencing experiment, we only used the animals received transplantation of hiPSC-derived microglia. We have made this clear in the revised manuscript.

Please also see the added information in Method (page 13, lines 40-48) and figure legends.

2. The data on “function” is not convincing. Although I don’t doubt the engrafted human cells are functional in the mouse brain, simply showing IHC for PSD-95 and Olig2 that colocalize with hTMEM119 or that human microglia are visualized near vascular structures is limited. Are there differences in these readouts between mouse and human microglia? Do human microglia respond to insult? Are they dynamic?

⇒ **Response:** Using the super-resolution imaging technique, we have performed new experiments to examine the functions of human Xeno MG and mouse microglia under homeostatic conditions. Furthermore, we compared their phagocytic functions under a toxin-induced demyelination condition.

Under homeostatic conditions, human Xeno MG and host mouse microglia similarly exhibited microglial functions, including synaptic pruning, phagocytosis of oligodendroglia, and having contact with blood vessels. As shown in the super-resolution images in **new Figure 3A, 3B and supplementary Figure 5A**, Xeno MG were found to engulf synaptic proteins at 8 weeks post-transplantation. Compared with human Xeno MG, we found fewer mouse microglia engulf synaptic proteins. To examine the function of phagocytosing oligodendroglia, in addition to OLIG2 staining, we further double-stained hCD45 with PDGFR α that is a marker for oligodendroglial progenitor cells. We observed that hCD45⁺ Xeno MG were able to engulf PDGFR α ⁺ oligodendroglia at 3 weeks post-transplantation in the corpus callosum (**new Figure 3D**). We also provided new merged zoom-in images as well as individual panels showing that hCD45⁺ Xeno MGs engulfed OLIG2⁺ oligodendroglia (**new supplementary Figure 6B**). In addition, we detected that a small population of the mouse microglia engulfed OLIG2⁺ oligodendroglia in the corpus callosum at 3 weeks post-transplantation (**new supplementary Figure 6C**). For the interaction with the blood vessels, we found that Xeno MG and mouse microglia similarly had close contact with blood vessels in the cerebral cortex, corpus callosum, and olfactory bulb (**Figure 3E and new supplementary Figure 6D and 6E**).

To explore whether Xeno MG are dynamic in response to insult, we fed the chimeric mice with cuprizone-containing diet to induce demyelination. The cuprizone model is one of the most frequently used models to study the pathophysiology of myelin loss in multiple sclerosis (Blakemore and Franklin, Curr Top Microbiol Immunol. 2008, PMID: 18219819). It is appropriate to use our hiPSC microglial chimeric mouse brain to examine the dynamics of human microglia under a demyelination condition, considering our observation that a large number of Xeno MG reside in the corpus callosum at 3 to 4 months post-transplantation and nearly only Xeno MG were found in the corpus callosum at 6 months post-transplantation (**new Figure 2C**). After 4 weeks of cuprizone treatment, we found that myelin structure, indicated by MBP staining in the corpus callosum was disrupted and became fragmented in our chimeric mice (**new Figure 6A**), in contrast to the intact and continuous MBP⁺ myelin structure in

chimeric mice fed with control diet (**new supplementary Figure 9**). As shown in the super-resolution images in **new Figures 6B and 6C**, engulfment of MBP⁺ myelin debris by Xeno MG and mouse microglia were clearly seen in the corpus callosum. Notably, more myelin debris was found inside of mouse microglia, compared with Xeno MG (**new Figures 6D**). In addition, we also examined the expression of CD74 and SPP1, which is known to be upregulated in multiple sclerosis (Masuda et al., Nature. 2019, PMID: 30760929). Without cuprizone treatment, variations in CD74 expression among animals were observed in Xeno MG and on average, about 20% of Xeno MG expressed CD74 (**new Figure 6E and F**). Nearly no Xeno MG expressed SPP1 in the corpus callosum (**new Figure 6G and H**). With cuprizone treatment, many of the Xeno MG expressed CD74 or SPP1, recapitulating the upregulated expression of CD74 and SPP1 in MS (**new Figures 6F**). Altogether, these results demonstrate human Xeno MG are dynamic in response to insult.

Please also see the added information in Results (page 6, lines 33-51; page 7, lines 2- 14; and page 10, lines 1-25).

3. Without sorting, it's unclear how the authors can be sure there are no human cells in other clusters (and vice versa- no mouse microglia within the human cluster?)

⇒ **Response:** The reviewer identifies an important point. We considered this issue carefully after the original submission and ended up posting a "monograph" on this topic on bioRxiv (<https://www.biorxiv.org/content/10.1101/671115v1>) to evaluate an optimized strategy for dealing with RNA-seq data from multiple species. Briefly, the final approach initially matched the entire scRNA-seq dataset to the mixed mouse-human reference genome files, and then chose, for each RNA-seq read ID, the optimally-aligned species using alignment scores stored in the BAM file. The original fastq files were then split into mouse-optimal and human-optimal subsets. Each of these was then re-mapped to a single species reference genome to eliminate any reduction in counts due to cross-species alignments. In the end, only 147 out of 19,154 barcodes, or 0.76%, included sequences optimally aligning with both species. These are likely caused by the creation of droplets containing cells from more than one species so these were eliminated from further consideration. Results from the manuscript have now been revised to use this strategy.

In order to minimize time between dissection and single-cell processing, we chose to skip FACS sorting since it would have added substantial processing time. A previous study has demonstrated that within hours during which microglia are isolated from the brain environment and transferred to culture conditions, microglia undergo significant changes in gene expression (Gosselin et al., Science. 2017, PMID: 28546318). Our goal was to capture observed expression patterns as close to the "in vivo" patterns as we could to the best of our abilities.

We have included this information in Results (page 7, lines 29-33) and Methods (page 15, lines 4-11).

4. Since mouse "microglia" potentially include perivascular macrophages and meningeal macrophages, how can the authors be sure that transcripts from these cell types are not contributing to the differences between human and mouse microglia (rather than the differences being based solely on the species of the cells)?

⇒ **Response:** We agree with this reviewer that mouse "microglia" potentially include perivascular macrophages and meningeal macrophages, but that this likely represents a small fraction of the clustered cells. Similarly, as shown in our **new supplementary Figure 3A-D** and a recent report

(Hasselmann et al., Neuron. 2019, PMID: 31375314), donor-derived cells in the mouse brain also included CNS macrophages, such as perivascular macrophages, choroid plexus macrophages, and meningeal macrophages. However, these CNS macrophages represents a small portion, because the vast majority of the hN⁺ donor-derived cells were hTMEM119⁺ microglia (around 93%, Fig. 1M). Therefore, each species of microglial cell cluster likely includes some sort of contaminating population. We couldn't exclude the possibility that these macrophages are involved. However, considering that those macrophages only have a small percentage compared with microglia, the major gene expression differences are likely to be driven by the majority microglia. This was also supported by the observation that previously-reported signature genes expressed in human microglia (Gosselin et al., Science. 2017, PMID: 28546318), and mouse microglia were differentially expressed in our sequencing data (**now Figure 5B and 5C**). Furthermore, we take this caution and only used the single-cell RNAseq to validate human origin of Xeno MG. We did not claim any novel gene expression related to differences between human vs. mouse microglia based on these data.

We have included this important point in the Discussion (page 12, lines 2-15).

5. In Fig. 4e it is very interesting data that enriched genes are related to innate immunity. However, could this be due to human cells in an otherwise mouse environment? For example, can human cells respond to other critical signals such as mouse Fractalkine, CD200, TGFb, etc.?

⇒ **Response:** The recent study (Hasselmann et al., Neuron. 2019, PMID: 31375314) elegantly demonstrated that murine CSF1 ligand is not enough to support the survival of engrafted human microglia and thus expression of humanized CSF1 in the recipient animals is both necessary and sufficient. However, the humanized genes such as hCSF2 and Thrombopoietin are not required, suggesting that the engrafted human microglia may receive those signals from murine counterpart molecules. Therefore, the finding of enriched innate immunity-related gene in our Xeno MG could be either the nature of human microglia as reported previously (Obstet Gynecol. 2013, PMID: 24150030; Galatro et al., Nat Neurosci. 2017, PMID: 28671693; Wolf et al., Annu Rev Physiol. 2017, PMID: 27959620), or the result from the responses to murine molecules, such as fractalkine, CD200, and TGFβ.

We have included this point in Results (page 9, lines 27-32).

Minor concerns:

1. In the introduction, lines 24-29 contains a very long sentence (“However, when cultured... from adult human brain tissue”). It may be easier for the reader to follow if split into two sentences. EX) “...hPSC-derived microglia best resemble fetal or early postnatal human microglia. This is indicated by...”

⇒ **Response:** We have split the long sentence into two sentences.

2. Page 5 line 21: “there are no changes in microglial number”

⇒ **Response:** Corrected.

3. Page 5 line 49: CD68 not “CD8”

⇒ **Response:** Corrected.

4. Page 21 line 30: “...from the mice that received transplantation...”

⇒ **Response:** Corrected.

5. Fig. 2a: Are inserts 1&2 from the proper location?

⇒ **Response:** We have indicated the correct location for the insert 2 and removed insert 1.

6. Fig. 2c-d: should be analyzed by a 2-way ANOVA since there are 2 independent variables: species and time.

⇒ **Response:** We agree with this reviewer's comment. We have re-analyzed the data and drawn the same conclusions. In the revised manuscript, we used two-way ANOVA to compare the endpoints and process length between human and mouse microglia, because there are two independent variables: species and time. For the comparison within mouse or human microglia, we used one-way ANOVA, because time was the only independent variable.

Please see the description in the legends of **new Figure 2D and 2E**.

7. Figure 2G: How many mice were used for each experiment? Are the images representative?

⇒ **Response:** There were 7 mice used for each experiment. To clarify, we quantified the percentage of CD68⁺ area in hTMEM119⁺ area (**now Fig. 2F**). In each Xeno MG, hTMEM119 staining usually labels the whole cell body and CD68, which is a marker for phagolysosomes, labels subregions of the cell body. The images in Fig. 2G are representative.

Reviewer #2:

Remarks to the Author:

In "Xenotransplantation of human PSC-derived microglia creates a chimeric mouse brain model that recapitulates features of adult human microglia," Xu and colleagues directly inject stem cell-derived human microglia (from embryonic or induced pluripotent stem cells) into immune immunocompromised murine hosts that express human CSF1, which is necessary for human microglial survival. They find that stem cell derived microglia-like cells engraft in the mouse brain and survive for long periods, terming them "Xeno MGs". They extensively characterize their model by examining engrafted cells at multiple timepoints, including after long term brain residence at 6 months, using mainly immunostaining and single cell RNAseq, across biological replicates. Their main conclusions are that Xeno MGs resemble human microglia at the levels of morphology, protein expression, function, and gene expression. They claim that Xeno Mgs model mature human microglia, as opposed in vitro stem cell derived Mgs, which are more similar to fetal microglia.

Stem cell-derived microglia-like cells have emerged as a tool to study human diseases, but suffer from the major limitation that microglia in vitro poorly resemble in vivo microglia, be they primary or stem cell-derived, from mouse or human. This made it very difficult to conclusively show whether stem-cell derived microglia were truly microglia. One group addressed this problem by injecting ips-microglia into the rodent brain, where they were shown to express microglia specific proteins (Blurton-Jones lab). But, whether they were true microglia remained incompletely resolved. The transplantation model described here significantly advances on these early studies by providing detailed characterization of transplanted human stem cell derived microglia-like cells, most importantly using single cell RNAseq to deeply

characterize cell identity. This is an important topic, with broad appeal.

The paper has many strengths, and has the potential to be an important asset to the field. Many parts are useful and well executed, including careful characterization of cell engraftment in figures 1 and 2, assessment of interspecies differences between host mouse microglia and transplanted xeno MGs at multiple timepoints in multiple biological replicates, and some treatment of whether the mouse brain environment can induce human microglial maturation. The manuscript is almost purely technical - it validates a new and very important model, but does not use this model to make new discoveries. Aside from this limitation of scope, it has three critical limitations that, if addressed, would significantly improve the manuscript.

⇒ **Response:** We greatly appreciate this reviewer's positive comments and constructive suggestions.

1. Analysis of transcriptomic data: broadly speaking, the authors have not analyzed their sequencing data in a way that compellingly demonstrates whether Xeno MGs closely resemble homeostatic human microglia.

*1A. Microglia signature genes: There is variation between mouse and human microglia, but many canonical microglial markers are conserved in human. Instead of focusing on these genes, the authors have overall chosen unusual genes and markers to argue for microglial identity. Many of these genes are expressed in microglia but are not microglia-specific, and some are more highly expressed in human compared to mouse microglia. Either way, these genes are not generally accepted as microglial identity markers, including in the human literature. It is critical to clearly show whether genes that indicate true microglial identity are expressed in Xeno MGs, and to validate these in tissue, which the authors have largely not done. In particular, it is important to 1) show expression of *Sall1* and a panel of other microglial identity markers (for example, *Slc2a5*, *P2ry12*, *Tmem119*, and *Olfml3*), 2) to quantify the percent of Xeno Mgs that express these genes, and 3) present the distribution of expression levels, compared to a control population such as pre-engraftment stem cell derived microglia.*

⇒ **Response:** In the revised manuscript, now we have first analyzed the common expression of microglial genes, including *C1QA*, *CX3CR1*, *TREM2*, *CSF1R*, and *P2RY12* (**new Figure 4C and 4D**), before choosing genes that distinguish human vs. mouse microglia (**new Figure 5B and 5C**). The results showed that these microglial genes were highly expressed in the clusters of Xeno MG and mouse microglia (**new Figure 4C and 4D**). Moreover, we have performed new bulk RNA-seq to analyze the pre-engraftment hiPSC-derived PMPs. In order to characterize microglial identity, we compared transcriptomic profiles between PMPs and Xeno MG from 6 months old chimeric mice. Notably, **as shown in new Figure 4E**, compared with PMPs, Xeno MG highly expressed microglial identity markers, such as *TMEM119*, *P2RY12*, *SALL1*, and *OLFML3*, which were barely detected in PMPs. Furthermore, we performed new immunostaining to validate the expression of *TMEM119* and *P2RY12* expression in PMPs and Xeno MG. Consistently, we did not detect any expression of *TMEM119* or *P2RY12* in PMPs (**new supplementary Figure 2D**). The vast majority of hN⁺ donor-derived cells similarly expressed *TMEM119* ($93.2 \pm 2.2\%$, $n = 7$, at 8 weeks post-transplantation; **Figure 1M**) and *P2RY12* ($93.4 \pm 3.8\%$, $n = 7$, at 6 months post-transplantation; **new supplementary Figure 2B and 2C**). Altogether, these results demonstrated that the mouse brain environment provides a conducive condition for transplanted hPSC-derived PMPs to develop and mature into microglia.

We also considered showing the expression of SALL1 at protein level by immunostaining. Unfortunately, none of the two SALL1 antibodies (Invitrogen, cat #: PA5-62057 and Everest Biotech, cat #: EB07779) we tested gave clean staining results (data not shown).

Please see added information in the Result (page 5, lines 8-11; and page 8, lines 8-13).

1B. Relatedly, in figure 3E, the authors argue that Xeno Mgs resemble microglia based on expression of a list of genes they claim is the top 30 expressed genes in primary human microglia from the Gosselin...Glass, Science 2017 paper. This list does not appear to be the same list as shown in the Glass paper (figure 1B of Glass paper), but even if it were, the most highly expressed genes in a cell type do not inherently define that cell's identity, and may overlap with many other cell types. In the Glass paper, this list was used to show inter-individual variability in highly expressed genes, not to characterize microglial identity.

⇒ **Response:** We thank this reviewer for pointing this out. We have removed the previous Figure 3E. In order to characterize microglial identity, as described in our above response to the comment #1A, we compared transcriptomic profiles between PMPs and Xeno MG and performed new immunostaining to examine the expression of TMEM119 and P2RY12. In addition, to further evaluate identity of Xeno MG in chimeric mouse brains, we compared the global expression patterns of 21 genes including the 11 microglia-specific genes and 7 HPC-specific genes shown in **new Figure 4F**, as well as 3 NPC genes (*NES*, *DCX*, and *SOX2*) between our Xeno MG, hiPSC-derived PMPs and published datasets of hiPSC-derived microglia cultured under 2-dimensional (2D) conditions (iPS MG) (Douvaras et al., Stem Cell Reports. 2017, PMID: 28528700), hiPSC-derived microglia developed in 3D cerebral organoids (oMG) (Ormel et al., Nat Commun. 2018, PMID: 30301888), hiPSC-derived microglia developed in mouse brain (xMG) reported in a recent study (Hasselmann et al., Neuron. 2019, PMID: 31375314). brain-tissue derived adult human microglia, including adult ex vivo microglia (Adult MG ExVivo) from Gosselin et al., 2017 (Gosselin et al., Science. 2017, PMID: 28546318) (age from 13-17 years) and Galatro et al., 2017 (age from 34-102 years) (Galatro et al., Nat Neurosci. 2017, PMID: 28671693), in vitro microglia (MG InVitro) from Gosselin et al., 2017 (Gosselin et al., Science. 2017, PMID: 28546318), as well as blood/liver macrophages (Douvaras et al., Stem Cell Reports. 2017, PMID: 28528700). As shown in **new Figure 4F**, a principal component analysis (PCA) demonstrated that Xeno MG together with xMG, were markedly distinct from blood/liver macrophages, PMPs, and the hiPSC-derived MG cultured under 2D conditions or developed in organoids. The human microglia cultured *in vitro* were separate from the other two clusters, which might suggest the significant impact of culture conditions on gene expression in those microglia as previously reported (Gosselin et al., Science. 2017, PMID: 28546318). Our Xeno MG clusters intermingled with a cluster of adult MG ExVivo samples collected from patient at ages from 13 to 102 years old, suggesting their resemblance to adult human microglia (**new Figure 4F**). All these results clearly indicate the microglia identity of our Xeno MG.

We have included this information in Results (page 8, lines 28-42).

1C. CD74 and SPP1 are not human microglia identity markers. In mouse they characterize white matter-associated subsets of microglia that appear during development (Li...Barres Neuron 2019, Hammond...Stevens Immunity 2019). In human, they are enriched in MS patient microglia (Masuda...Prinz Nature 2019). Either way, they do not specifically mark microglia as opposed other macrophages and so do not attest to xeno Mgs being microglia-like. Furthermore, although mRNA appears to be present in the Masuda paper in homeostatic microglia, virtually no microglia express Spp1 and very few express CD74 in the healthy human brain by immunostaining. Thus, if anything, high

expression of these markers suggests non-homeostatic microglia, which should be clarified, for example by tissue staining to compare to published results, and comparative analysis of expression levels in homeostatic microglia from the Masuda paper to Xeno MGs.

⇒ **Response:** We agree with this reviewer's comment. In the revised manuscript, we have shown that the highly abundant expression of *CD74* and *SPP1* in Xeno MG cluster distinguished it from mouse microglia cluster (**new Figure 5B and C**). To further examine *CD74* and *SPP1* expression at protein level in Xeno MG under homeostatic conditions, we double-stained *CD74* or *SPP1* with hN or hCD45, respectively. We found that about 20% of Xeno MG expressed *CD74* and nearly no Xeno MG expressed *SPP1* in the corpus callosum at 3 months post-transplantation (**new Figure 6E-6H**). This is in line with the findings in Masuda paper. We also think that the discrepancy between the *CD74* and *SPP1* transcript levels and their protein levels may suggest an involvement of sophisticated control of mRNA translation that was observed in regulation of other genes critical for brain development (Kwan et al., Cell. 2012, PMID: 22579290; Kraushar et al., Proc Natl Acad Sci U S A. 2014, PMID: 25157170).

As this reviewer also pointed out, *CD74* and *SPP1* are highly expressed in MS patients and animal models of MS (Masuda et al., Nature. 2019, PMID: 30760929). Therefore, we fed our chimeric mice with cuprizone-containing diet to induce demyelination. The cuprizone model is one of the most frequently used models to study the pathophysiology of myelin loss in MS (Blakemore and Franklin, Curr Top Microbiol Immunol. 2008, PMID: 18219819). After 4 weeks of cuprizone treatment, many of the Xeno MG expressed *CD74* or *SPP1* (**new Figures 6E and F**), recapitulating the upregulated expression of *CD74* and *SPP1* in MS. These results suggest that our Xeno MG likely represent homeostatic human microglia under homeostatic conditions and are also dynamic, in response to insult.

We have included this information in Results (page 10, lines 14-25), and in Discussion (page 12, lines 20-24).

1D. To show that the mouse CNS environment induces Xeno MG maturation as claimed, the authors would need to compare a very early timepoint after engraftment to a long term timepoint. This is an important piece of data for a resource paper that may form the basis for experiments in other labs predicated on this claim. Furthermore, aging is not the same as maturation from a fetal-like state to an adult state. Thus, validation of maturation using an aging signature does not show maturation.

⇒ **Response:** We agree with this reviewer that it would be greatly helpful if we could obtain the transcriptomic profile of Xeno MG at a very early time point. However, there were much fewer Xeno MG in chimeric mouse brain at very early time points than the brain at a long-term time point, as exemplified in the olfactory bulb in week 3 brain vs. month 6 brain (**Figure 1 E and F**). Technically, it is extremely challenging to collect enough number of Xeno MG at early time points for reliable scRNA-seq analysis.

As an alternative approach, we have performed new bulk RNA-seq to analyze the pre-engraftment hiPSC-derived PMPs, as also described in our above response to the comment #1A. We compared transcriptomic profiles between PMPs and Xeno MG from 6 months old chimeric mice. Notably, **as shown in new Figure 4E**, compared with PMPs, Xeno MG highly expressed microglial identity markers, such as *TMEM119*, *P2RY12*, *SALL1*, and *OLFML3*, which were barely detected in PMPs. On the other hand, the expression of markers for hematopoietic progenitor cells, such as *CD59*, *CD44*, and *CD38*, was low in Xeno MG but much higher in PMPs. Furthermore, we performed new immunostaining to validate the expression of *TMEM119* and *P2RY12* expression in PMPs and Xeno MG. Consistently, we did not

detect any expression of TMEM119 or P2RY12 in PMPs (**new supplementary Figure 2D**). The vast majority of hN⁺ donor-derived cells similarly expressed TMEM119 (93.2 ± 2.2%, n = 7, at 8 weeks post-transplantation; **Figure 1M**) and P2RY12 (93.4 ± 3.8%, n = 7, at 6 months post-transplantation; **new supplementary Figure 2B and 2C**). These results demonstrate that the mouse brain environment provides a conducive condition for transplanted hPSC-derived PMPs to develop and mature into microglia. We have included the new information in Results (page 5, lines 8-11; and page 8, lines 8-13).

We also agree with this reviewer's comment on aging gene list and have removed that PCA plot.

1E. Relatedly, it is critical to see what stem cell-derived microglia-like cells look like transcriptomically prior to engraftment in the brain. This is related to point 1D, and will again allow readers to understand the effects of the brain environment on Xeno Mgs.

⇒ **Response:** Please see our above response to the comment #1D.

1F. Figure 3F - how are the authors controlling for batch effects? It is not clear how downsampling reads addresses this. It is important to explain methods used to test for, and/or adjust for batch effects, which could partially or fully explain their clustering.

⇒ **Response:** We thank the reviewer for bringing up this important point. In our original study, we attempted to balance comparisons between very deep datasets and our relatively shallow human microglia data by randomly sampling from the deep data. The intent was to prevent inflation of low-quality counts in the shallower dataset. But upon further consideration, we tested several alternate approaches. We now use the full dataset from each sample and rely on both the modeling in DESeq2 with the data source being a factor and the subsequent variance-stabilized log transformation (vst) to account for batch effects. This approach, which was recommended by Mike Love, the author of DESeq2, models the batches to produce residuals with comparable medians and ranges (<https://mikelove.github.io/pages/software.html>). Extending this to include the limma removeBatchEffects function tended to compress data to a much tighter range, eliminating most differences between samples. For example, PCA analysis of removeBatchEffects data tightly grouped human adult microglia with blood and liver macrophages (not shown). Again, we checked with posts from Mike Love and he argued against adding the removeBatchEffects step to a DESeq2/vst data matrix. Simple barplots demonstrate these observations. We chose to use DESeq2/vst modeling and transformation, without prior downsampling, for the PCA study.

We have included this information in the Method details (page 15, lines 15-20).

2. Synapse engulfment by Xeno MGs is not validated. The authors show PSD95 positive puncta that appear to be inside Xeno MGs in an image reconstruction. They interpret this to mean that Xeno MGs prune synapses in the rodent brain. The inclusion of functional data is appreciated and really strengthens the paper. The finding that human macrophages can engulf mouse synapses, however, is a major finding. It would be the first validation that human microglia-like cells perform microglia-specific functions in vivo. As such, concluding that Xeno MGs engulf synapses requires further validation of the presented imaging data. For one, staining with a post-synaptic marker does not equate to synapse engulfment, which requires colocalized pre-and post synaptic marker staining. Second, confocal imaging alone is insufficient to prove engulfment. This requires a high resolution imaging technique (for example EM, array tomography, possibly super-resolution imaging) to conclusively show synaptic puncta are inside microglia.

⇒ **Response:** To validate synaptic pruning function of Xeno MG in mouse brain, we have performed the following new experiments and generated new data. First, we employed the super-resolution imaging technique to better visualize synapse engulfment by Xeno MG. In **new Figure 3A**, the clear engulfment of synaptic proteins by Xeno MG is demonstrated by zoom-in images with individual panels as well as merged images. Second, we triple-stained hTMEM119 with both a post-synaptic marker PSD95 and a pre-synaptic marker synapsin I. The new 3D reconstruction images show that PSD95⁺ and synapsin I⁺ puncta are colocalized within hTMEM119⁺ processes, indicating that these synaptic proteins are phagocytosed by Xeno MG at eight weeks post-transplantation in grey matter (**new Figure 3A**). In addition, we also validated the specificity of PSD95 puncta staining by incubating brain sections with the PSD95 antibody together with a PSD95 peptide. We barely detected any PSD95⁺ puncta signal after the incubation in the presence of PSD95 peptide (**new supplementary Figure 5A**). Lastly, we also triple-staining hTMEM119 and PSD95 with CD68, a marker of phagolysosomes (DeFalco et al., Proc Natl Acad Sci U S A. 2014, PMID: 24912173). As shown in **new Figure 3B and new supplementary Figures 5B, C**, PSD95⁺ puncta are localized within CD68⁺ phagolysosomes in hTMEM119⁺ Xeno MG, further indicating the synaptic pruning function of Xeno MG. Of note, this engulfment of synaptic materials was observed from 3 weeks to 6 months post-transplantation, with a peak at 8 weeks post-transplantation (**new Figure 3C**). Taken together, all these results demonstrate that human Xeno MG can prune synapses in mouse brain.

We have also added this information in Results in the revised manuscript. Please see Results (page 6, lines 33-47).

3. The results section frequently editorializes about what presented findings might mean, in a way that is misleading. The manuscript is carefully written and makes many good points, but also strongly asserts what findings may mean in the results section. Usually, the authors do qualify these statements so that technically speaking they are not making inappropriate conclusions and rather are speculating, but this reviewer still found these implied conclusions to read like overstatements. Below are some examples, but overall, these unsubstantiated speculations detract from the text. Many of them could be toned down and included in the discussion.

3A. Pg 4 line 35-6 - "highly ramified morphology typical of resting microglia," (or similarly Pg 7 lines 11-12). The concept of "resting" microglia is poorly defined and non-specific, and although microglia do change morphology in some contexts, a ramified morphology does not mean the cells are "resting."

⇒ **Response:** We have removed the descriptions about "resting" microglia.

page 5, lines 27-31 - the authors use words like expel and replace to describe Xeno Mg effects on host microglia - they have not shown this, and there are many explanations for observed patterns of engraftment.

⇒ **Response:** We have removed that description.

3B. Page 8, lines 20-21 - the authors state that human microglia may be more immunocompetent than mouse - this is not addressed in the data

⇒ **Response:** We have removed that statement and changed to “Xeno MG and mouse microglia exhibited similar overall patterns of transcriptomic profile, but numerous species-specific differentially expressed genes were also observed.” Please see page 9, lines 29-32.

3C. page 4 lines 22-43 - this paragraph has a large amount of speculation about why microglia end up where they do after injection, that is not substantiated by data in the paper.

⇒ **Response:** We have removed most of these descriptions. In addition, we performed new transplantation experiments to support the speculation. In the new experiments, we deposited hiPSC-derived PMPs into different sites, the lateral ventricles of P0 mice. At three weeks post-transplantation, we found that the majority of donor-derived cells migrated along the anterior corpus callosum, rostral migration stream, and then resided in the olfactory bulb. Moreover, some of those cells of migrated posteriorly along the corpus callosum (**supplementary Figure 1B**). As shown in our previous results, when deposited into the anlagen of the corpus callosum above the hippocampus, donor-derived hTMEM119⁺ microglia also migrated long distances along the corpus callosum to reach the olfactory bulb (**Figure 1D**). In addition, previous studies have shown that microglia use vessels and white matter tracts as guiding structures for migration and enter all brain regions (Kettenmann et al., *Physiol Rev.* 2011, PMID: 21527731). We propose that the engrafted human microglia likely used corpus callosum to migrate to various brain regions.

Please see added information in Results (page 4, lines 45-50).

3D. Page 4, lines 14-14 “PMPs are produced in a Myb-independent manner that closely recapitulated primitive hematopoiesis” - has the protocol used by this group truly been proven to be myb independent? If so, cite.

⇒ **Response:** The iPSC-derived PMPs generated using this protocol have been proven to recapitulate myb-independent primitive hematopoiesis in the previous studies (Buchrieser et al., *Stem Cell Reports.* 2017, PMID: 28111278; Haenseler et al., *Stem Cell Reports.* 2017, PMID: 28591653). We have cited the original research articles in the revised manuscript.

3E. Page 7, lines 12-19 - lack of inflammatory cytokine production speaks to certain kinds of inflammatory microglial reactivity, but does not strongly argue that the cells are “non activated” (a very broad and very nonspecific term).

⇒ **Response:** We have removed that statement.

Minor points:

1. Page 6, lines 8-9. Microglia are not commonly considered a component of the BBB. Still, it is fine to say this if a citation is provided.

⇒ **Response:** We have added the citations (Zlokovic, *Neuron.* 2008, PMID: 18215617; da Fonseca et al., *Front Cell Neurosci.* 2014, PMID: 25404894; Dudvarski Stankovic et al., *Acta Neuropathol.* 2016, PMID: 26711460).

2. Page 6, lines 32-33 - the authors state that FACS has the potential to impact gene profiles through extended ex vivo manipulation. It is more likely that enzymatic digestions at warm temperatures cause

ex vivo activation, which the authors do employ - the impacts of FACS have to my knowledge been specifically tested in isolation.

⇒ **Response:** We have removed this statement about FACS and replaced with “A previous study has demonstrated that within hours during which microglia are isolated from the brain environment and transferred to culture conditions, microglia undergo significant changes in gene expression (Gosselin et al., Science. 2017, PMID: 28546318). To capture observed expression patterns as close to the “in vivo” patterns as possible, we chose to omit a FACS sorting step since it would have added substantial processing time”. Please see page 7, lines 29-33.

3. For *ki67* staining of human cells, it would be helpful to also show the levels of staining in mouse cells.

⇒ **Response:** We have double-staining mTMEM119 with Ki67. Then, we examined different brain regions at 3 weeks post-transplantation. We only found a very small number of mouse microglia express Ki67 in the subventricular zone (**new supplementary Figure 3F**), which is consistent with the previous report (Raj et al., Neurobiol Aging. 2014, PMID: 24799273).

Page 5, line 49 - typo (CD8 → CD68)

⇒ **Response:** Corrected.

4. Figure 1 - what are the TMEM119 negative human cells? Are they macrophages, as evidenced by IBA1 staining? If not, the authors should comment on what they are.

⇒ **Response:** We have performed new immunostaining to characterize the identity of hTMEM119 negative cells. As suggested by this reviewer’s #5 comment below, we examined the distribution of human donor cells in border regions, including choroid plexus, meninges, and perivascular spaces. We found that most of the hTMEM119⁻/human nuclei (hN)⁺ donor-derived cells were seen in those border regions. Furthermore, we co-stained hTMEM119 with CD163, an established marker for non-microglial CNS myeloid cells (Goldmann et al., Nat Immunol. 2016, PMID: 27135602; Hasselmann et al., Neuron. 2019, PMID: 31375314). In choroid plexus, we found that some of the hTMEM119⁻ cells co-expressed CD163, suggesting that these transplanted cells differentiated into choroid plexus macrophage (cpMΦ), but not microglia (**new supplementary Figure 3A**). In order to better visualize meninges and perivascular space, we triple-stained hN and CD163 with laminin, a marker that has been commonly used to visualize vascular structures in the mammalian brain (Eriksdotter-Nilsson et al., J Neurosci Methods. 1986, PMID: 3537540). There was also a small number of hN⁺ and CD163⁺ co-expressing cells in these regions, suggesting that the transplanted cells differentiated into meningeal macrophage (mMΦ) and perivascular macrophages (pvMΦ) (**new supplementary Figure 3B, 3C**).

In addition, there was a possibility that some transplanted cells might remain as progenitors and maintain their hematopoietic progenitor-like cell identity. We thus stained CD235, a marker for yolk sac primitive hematopoietic progenitors. As shown in **new supplementary Figure 3D**, we did find that a small population of hN⁺ cells expressed CD235 in the regions close to the lateral ventricles. Overall, these results demonstrate that the vast majority of engrafted hiPSC-derived PMPs differentiate into hTMEM119⁺ microglia, with a relatively small number giving rise to other of hTMEM119⁻ CNS myeloid cells in a brain context-dependent manner or remaining as progenitors.

We have included this information in the revised manuscript. Please see Results (page 5, lines 11-27).

5. Do the authors observe engraftment of xeno MGs in border regions, eg meninges, choroid, perivascular spaces? Do they find any donor cells outside the brain (particularly in the blood?) These are useful additions for a resource paper.

⇒ **Response:** We did observe donor-derived cells in the border regions, which are likely to be non-microglial CNS myeloid cells (**new supplementary Figure 3**). Please also see our above response to the comment # 4.

We did not find any donor cells outside of the brain. However, we cannot exclude the possibility that the donor cells may migrate to the blood, because our immunostaining was performed after transcatheter perfusion to remove blood. It will be interesting to examine this possibility in future studies.

6. Page 4, lines 14-15: the authors claim that their Xeno MGs come from myb independent cells. Was this proven, or speculative?

⇒ **Response:** We did not examine Myb expression in our Xeno MG. However, the protocol we used to generate PMPs from hPSCs has been proven to recapitulate myb independent primitive hematopoiesis in the previous studies (Buchrieser et al., Stem Cell Reports. 2017, PMID: 28111278; Haenseler et al., Stem Cell Reports. 2017, PMID: 28591653). We have cited the original research articles in the revised manuscript.

Reviewer #3:

Remarks to the Author:

The manuscript from Xu et al. provides the description of a novel model to create mouse-human chimera with human microglia engrafted into the brains of mice. The authors demonstrate that primitive hematopoietic progenitors transplanted into neonatal Rag2KO/il2rgKO/hCSF1 mice are capable of engrafting through many regions of the brain and express many of the canonical microglia markers at both the protein and gene levels. The authors also begin the process of examining some of the differences between human and mouse microglia within their model. While this could represent a step forward in a new model system that would allow for readouts of human microglia in a more in vivo-like setting there is not a demonstration that this is the case. The key missing aspect is some sort of challenge and examination of the functional differences in response between a WT human cell line and an isogenic cell line with a key functional microglial gene knocked out (Trem2, P2ry12, etc). Without this more meaningful functional data this paper may be better served as a methods/technique paper.

⇒ **Response:** We thank this reviewer for the favorable comments and critical suggestions.

Major Critiques

1. *The paper and discussion section proposes some important goals for this model but there is not proof of concept that they can be effectively studied in mouse chimera system or specific useful endpoints examined. All the work presented was done at homeostatic conditions. In order to state that this could be useful model to examine human microglia and relevance to disease, genetic background, environment, etc there is a need to show that such changes can be detected and functional readout in this model.*

--The best way to accomplish this would be to perform transplantation experiment where some mice are transplanted with WT HPCs and another set are transplanted with isogenic cell line with key functional

microglial gene knocked out (i.e. Trem2, P2ry12, etc). Then challenge the mice in relevant fashion (disease model, infection, etc) and demonstrate that an altered response can be detected in the KO cell line. While this is extreme case vs comparing different patient lines to each other it would provide some proof of concept to the ability of this model to readout functional changes.

⇒ **Response:** While our manuscript was under review, a paper was published in Neuron (Hasselmann et al., Neuron. 2019, PMID: 31375314), reporting the generation of a chimeric mouse brain that is very similar to ours. In that study, hiPSC-derived microglia carrying wild-type TREM2 or R47H mutant were examined in a context of Alzheimer's disease.

To distinguish our study for the published report, we examined the dynamic responses of our Xeno MG under a cuprizone-induced demyelination condition. The cuprizone model is one of the most frequently used models to study the pathophysiology of myelin loss in multiple sclerosis (Blakemore and Franklin, Curr Top Microbiol Immunol. 2008, PMID: 18219819). It is appropriate to use our hiPSC microglial chimeric mouse brain to examine the dynamics of human microglia under a demyelination condition, considering our observation that a large number of Xeno MG reside in the white matter corpus callosum at 3 to 4 months post-transplantation and nearly only Xeno MG were found in the corpus callosum at 6 months post-transplantation (**new Figure 2C**). After 4 weeks of cuprizone treatment, we found that myelin structure, indicated by MBP staining in the corpus callosum was disrupted and became fragmented in our chimeric mice (**new Figure 6A**), in contrast to the intact and continuous MBP⁺ myelin structure in chimeric mice fed with control diet (**new supplementary Figure 9**). As shown in the super-resolution images in **new Figure 6B and 6C**, engulfment of MBP⁺ myelin debris by Xeno MG and mouse microglia were clearly seen in the corpus callosum. Notably, more myelin debris was found inside of mouse microglia, compared with Xeno MG (**new Figure 6D**). In addition, we also examined the expression of CD74 and SPP1, which is known to be upregulated in multiple sclerosis (Masuda et al., Nature. 2019, PMID: 30760929). Without cuprizone treatment, variations in CD74 expression among animals were observed in Xeno MG and on average, about 20% of Xeno MG expressed CD74 (**new Figure 6E and F**). Nearly no Xeno MG expressed SPP1 in the corpus callosum (**new Figure 6G and H**). With cuprizone treatment, many of the Xeno MG expressed CD74 or SPP1, recapitulating the upregulated expression of CD74 and SPP1 in MS (**new Figures 6F**). Altogether, these new results demonstrate that human Xeno MG are dynamic in response to insult and provide proof of concept that our chimeric mouse model has the ability to read out functional changes of human microglia under disease conditions.

We have included this information in the revised manuscript. Please see Results (page 10, lines 1-25).

2. Any chimeric model is going to have a question regarding its reproducibility and variance. The authors have done an admirable job of several aspects to address this but some relatively minor but important analyses should be performed/presented. As minor note displaying all data points overlaid on bar graphs significantly aids in interpretation of data and should be added.

a. Please include SI figure with whole tile scan (same as in Figure 1C) with several different animals next to each other to visualize differences in engraftment. While the authors present data on transcriptome and cell number examining regional differences in engraftment between animals is also key. Should be matter of presenting images used in quantifications of Figure 1L.

⇒ **Response:** We thank this reviewer for bringing forward this important point. We now have displayed all data points overlaid on bar graphs for all of the quantification data throughout the figures. As shown

in **new Figure 1L**, there was indeed variations in chimerization among animals, as indicated by the percentage of hTMEM119⁺ Xeno MG in total DAPI⁺ cells in the forebrain. Therefore, we included new tile scan images collected from another chimeric mouse brain that represent the lower level of chimerization (**new supplementary Figure 1A**).

On the other hand, this model is also highly reproducible, according to the scRNA-seq analysis using four chimeric mouse brains. We calculated the numbers of detected mouse/human microglia in each mouse brain sample and found that human microglia were consistently detected in each sample. The numbers (with percentages in parentheses) of detected mouse/human microglia, respectively, detected in each sample were: Sample 1: 442/395 (21.6%/19.3%); Sample 2: 176/274 (18.0%/28.1%); Sample 3: 551/537 (14.6%/14.3%); Sample 4: 383/327 (17.2%/14.7%). In addition, the high reproducibility of generating such a hiPSC microglial chimeric mouse brain model was also corroborated by two other recent reports (Hasselmann et al., Neuron. 2019, PMID: 31375314; Mancuso et al., bioRxiv <https://doi.org/10.1101/562561>. 2019).

We have added this information in Results (page 4, lines 39-43) and Discussion (page 11, lines 33-38).

b. It is important for the authors to not only provide the percent of human cells out of all DAPI⁺ cells (Figure 1L) but also the percentage of microglia that are hTMEM119⁺ (ideally broken down by different brain regions)

⇒ **Response:** We have quantified the percentage of hTMEM119⁺ microglia in total microglia (hTMEM119⁺ cells plus mTMEM119⁺ cells) in the cerebral cortex, hippocampus, and corpus callosum at 6 months post-transplantation. Please see the **new Figure 2C**.

c. The authors need to address what markers are expressed by the remaining 5-10% of cells (Figure 1M) that are TMEM119⁻. Are these cells Ki67⁺? PU.1?, Do they still have expression of HPC markers? Also should address if these cells are more common in any given brain region or are spread throughout the brain.

⇒ **Response:** We have performed new immunostaining to characterize the identity of hTMEM119 negative cells. We further examined the distribution of human donor cells in border regions, including choroid plexus, meninges, and perivascular spaces. We found that most of the hTMEM119⁻/human nuclei (hN)⁺ donor-derived cells were seen in those border regions. Moreover, we co-stained hTMEM119 with CD163, an established marker for non-microglial CNS myeloid cells (Goldmann, et al., Nat Immunol. 2016, PMID: 27135602; Hasselmann et al., Neuron. 2019, PMID: 31375314). In choroid plexus, we found that some of the hTMEM119⁻ cells co-expressed CD163, suggesting that these transplanted cells differentiated into choroid plexus macrophage (cpMΦ), but not microglia (**new supplementary Figure 3A**). In order to better visualize meninges and perivascular space, we triple-stained hN and CD163 with laminin, a marker that has been commonly used to visualize vascular structures in the mammalian brain (Eriksdotter-Nilsson, et al., J Neurosci Methods. 2016, PMID: 3537540). There was also a small number of hN⁺ and CD163⁺ co-expressing cells in these regions, suggesting that the transplanted cells differentiated into meningeal macrophage (mMΦ) and perivascular macrophages (pvMΦ) (**new supplementary Figure 3B, 3C**).

In addition, we stained CD235, a marker for yolk sac primitive hematopoietic progenitors, to examine if transplanted cells remained as progenitors and maintained their hematopoietic progenitor-like cell identity. As shown in **new supplementary Figure 3D**, we did find that a small population of hN⁺ cells

expressed CD235 in the regions close to the lateral ventricles. Overall, these results demonstrate that the vast majority of engrated hiPSC-derived PMPs differentiate into hTMEM119⁺ microglia, with a relatively small number giving rise to other of hTMEM119⁻ CNS myeloid cells in a brain context-dependent manner or remaining as progenitors.

We have included this information in the revised manuscript. Please see Results (page 5, lines 11-27).

3. As a paper presenting a model/resource for future use by the scientific community, the methods need to very detailed and thorough. For this reason, many small issues with methods details have been elevated to major critique and requires significantly increased detail to be included. Some examples of information missing includes:

--What passage were iPSC and ESC lines before starting differentiation?

⇒ **Response:** The passage numbers of the hiPSC and hESC lines were between P20 to P30. Please see the added information in Methods (page 13, lines 11-12).

--What are the sexes of the donors?

⇒ **Response:** Both hiPSC and ESC lines are from female donors. Please see the added information in the Method (page 13, line 4).

--Specify which 10X single cell gene expression kit was used for capture and library preparation?

⇒ **Response:** We have used Chromium™ i7 Multiplex Kit, 96 rxns, Chromium™ Single Cell 3' Library and Gel Bead Kit v2 and Chromium™ Single Cell A Chip Kit for capture and library preparation. The methods section description of these steps has been expanded. Please see the added information in Methods (page 14, line 31-33).

--Significantly more details are needed for the single cell analysis pipeline. Stating loading of data into Seurat is not sufficient. Details need to be given about both QC of low quality cells and the numerous choices (variables regressed, number of PCs, clustering resolution, etc) during processing that can have dramatic effects on resulting analysis.

--The results text states stringent criteria were used for QC but no information is provided as to what that criteria was.

⇒ **Response:** We have posted detailed bioinformatic methods behind our data analysis pipeline in bioRxiv (<https://doi.org/10.1101/671115>), including all R commands, python scripts, data quality evaluation, and parameters used. In the revised manuscript, this posting is cited so as not to distract from the biological interpretation of results with these technical details.

Please see the added information in Method (page 15, lines 4-11).

--Significantly more details of IHC and imaging protocols and analysis are also needed

⇒ **Response:** We have provided more details of IHC and imaging protocols and analysis in the Method (page 15, lines 23-51).

4. There are couple issues with single cell sequencing that should be remedied to strengthen the paper. First and foremost is that the authors need to examine how the presence of human microglia may modulate other CNS cell types. Without an understanding of this it is not possible to fully evaluate the utility of the model.

--To accomplish this the authors should perform transplant study with mouse microglia (to mimic surgery and foreign cell transplant) and then perform the same single cell sequencing. This will allow for comparisons of gene expression from non-microglial cell types that are included in the dataset.

⇒ **Response:** We agree with this reviewer that this would be an important study, but it's clearly outside the scope of this manuscript. Our goal here was to examine expression patterns and functions of the transplanted human microglia. We chose to use whole brain sections to minimize handling time prior to cell lysis, so we included our results from the host mouse brain cells and utilized the clustered mouse brain cells as a contrast to human Xeno MG, but examining the impact of Xeno MG on mouse brain cells was not the primary focus of this study.

However, we thank the reviewer for this excellent idea, because a previous study did report that engrafted human astrocytes modulated other CNS cell types in the mouse brain, particularly enhancing neuronal synaptic plasticity (Han et al., Cell Stem Cell. 2013, PMID: 23472873). In future experiment, it would be fascinating to examine the impact of engrafted human microglia on neuronal development, synaptic plasticity, as well as the impact on behavioral performance of the animals. We have included this point in and Discussion (page 12, lines 11-15).

Other issues with single cell can be remedied with new or altered analyses of existing data:

a. Single cell sequencing is relatively shallow (~15,000 and may not be optimally powered to do full differential expression between cell types).

⇒ **Response:** We agree with this comment, but the relatively shallow scRNA-seq data is sufficient for validating the identity of our human iPSC-derived microglia after transplant. In the revised manuscript, we now focused on the use of single-cell RNA-seq to validate the identity of our Xeno MG against multiple, previously-published, much deeper datasets. Our intent was not to repeat previously published studies of human vs. mouse microglial expression patterns. As one good example, Galatro et al. (2017) and Goseelin et al. (2017) did a very careful mouse vs. human study (Galatro et al., Nat Neurosci. 2017, PMID: 28671693; Gosselin et al., Science. 2017, PMID: 28546318). In addition, this caution is well taken and we did not claim any novel gene expression based on these data.

b. Several statistics should be reported that are not currently. These statistics are important for the interpretation of differential expression and gene expression data that is presented.

--Genes & UMIs detected per cell (mouse microglia)

--Genes & UMIs detected per cell (human microglia)

--Number of mouse microglia, human microglia, and ratio for each of the individual samples.

⇒ **Response:** The summary of numbers of sequencing reads, barcodes, and mapping to genome was already presented in the table in **new supplementary Figure 7A**. The numbers of detected mouse and human microglia are part of the pie chart in **new supplementary Figure 7D**. The legend to **supplementary Figure 7** now includes the ratio of mouse and human microglia for each individual sample: The numbers (with percentages in parentheses) of detected mouse/human microglia, respectively, detected in each sample were: Sample 1: 442/395 (21.6%/19.3%); Sample 2: 176/274

(18.0%/28.1%); Sample 3: 551/537 (14.6%/14.3%); Sample 4: 383/327 (17.2%/14.7%). However, it should be clear that these numbers do not represent percentages of the entire brain, only of the region we dissected as described in the methods. This has now been made clear in our descriptions. Please see page 8, line 3.

c. The table presented in S3A line entitled “total” is misleading and should be changed to averages. The sum totals of those values are not relevant but averages are important.

⇒ **Response:** “Total” has been changed to “averages” where appropriate in this table. Please see now **supplementary Figure 7A**.

d. The statement that clustering pattern was consistently observed needs to be clarified or rephrased. --As current worded it implies that if each animal was analyzed on it’s own that the same 11 clusters were present. However, it is not clear from Figure S3B whether those plots are representative of individual analyses from each animal or whether they represent the contribution of each to the clustering results when all animals are analyzed together. That is key difference and if the later is true the text needs to be rephrased as to not be misleading.

⇒ **Response:** To address this concern, results from each replicate animal were analyzed separately and the resulting tSNE clustering consistently identifies the same 11 clusters. This was not shown in the original submission but is now included as **new supplementary Figure 7B**.

5. Phagocytosis data is not entirely convincing.

a. For PSD95 all individual panels should be presented in panel and not just merged image in IMARIS representation. Additionally, images should be stained using CD68 as well to confirm phagocytosis. Furthermore authors should consider using antibody that is KO validated as punctate synaptic staining can be difficult to properly interpret.

⇒ **Response:** We greatly appreciate the suggestions from this reviewer. To confirm synaptic pruning function of Xeno MG in mouse brain, we have performed the following new experiments and generated new data. First, we employed the super-resolution imaging technique to better visualize synapse engulfment by Xeno MG. In **new Figure 3A**, the clear engulfment of synaptic proteins by Xeno MG is demonstrated by zoom-in images with individual panels as well as merged images. Second, we triple-stained hTMEM119 with both a post-synaptic marker PSD95 and a pre-synaptic marker synapsin I. The new 3D reconstruction images showed that PSD95⁺ and synapsin I⁺ puncta were colocalized within hTMEM119⁺ processes, indicating that these synaptic proteins are phagocytosed by Xeno MG at eight weeks post-transplantation in grey matter (**new Figure 3A**). Lastly, we also triple-staining hTMEM119 and PSD95 with CD68, a marker of phagolysosomes (DeFalco et al., Proc Natl Acad Sci U S A. 2014, PMID: 24912173). As shown in **new Figure 3B and new supplementary Figures 6B, C**, PSD95⁺ puncta were localized within CD68⁺ phagolysosomes in hTMEM119⁺ Xeno MG, further indicating the synaptic pruning function of Xeno MG. Of note, this engulfment of synaptic materials was observed from 3 weeks to 6 months post-transplantation, with a peak at 8 weeks post-transplantation (**new Figure 3C**). Taken together, all these results demonstrate that human Xeno MG can prune synapses in mouse brain.

We have added this information in Results (page 6, lines 33-47).

b. The images and data of Olig2 engraftment is very poor. Neither of the two images are at all convincing that the Olig2 nuclear is inside the microglia cell body. Much better images are needed and again all

individual changes should be presented and not just a merged tilted projection (i.e. Figure S2B top panel).

⇒ **Response:** We have provided new representative images with separate and merged channels to show that hCD45⁺ microglia phagocytose Olig2⁺ oligodendroglia (**new supplementary Figure 6B**). In addition, to further confirm that, we double-stained hCD45 with another oligodendroglial marker PDGFR α . The results clearly show that hPSCs-derived microglia in white matter engulfed PDGFR α ⁺ cell body at 3 weeks post-transplantation (**new Figure 3D**).

We have also described this in Results (page 6, lines 48-51).

6. There are couple overstatements that need to be adjusted.

--Stating microglia use white matter to migrate and enter brain regions when cells were transplanted into the white matter is little bit circular reasoning.

--To make this statement one would need to see if this pattern is observed if cells are transplanted elsewhere and then use white matter to migrate

⇒ **Response:** We have revised the statement to that “the engrafted human microglia likely used corpus callosum to migrate to various brain regions”. Moreover, we performed new transplantation experiments to support this idea. In the new experiments, we deposited hiPSC-derived PMPs into different sites, the lateral ventricles of P0 mice. At three weeks post-transplantation, we found that the majority of donor-derived cells migrated along the anterior corpus callosum, rostral migration stream, and then entered the olfactory bulb. Moreover, some of those cells of migrated posteriorly along the corpus callosum (**supplementary Figure 1B**). As shown in our previous results, when deposited into the anlagen of the corpus callosum above the hippocampus, donor-derived hTMEM119⁺ microglia also migrated long distances along the corpus callosum to reach the olfactory bulb (**Figure 1D**). In addition, previous studies have shown that microglia use vessels and white matter tracts as guiding structures for migration and enter all brain regions (Kettenmann et al., *Physiol Rev.* 2011, PMID: 21527731). Therefore, we propose that the engrafted human microglia likely used corpus callosum to migrate to various brain regions

We have added this information in Results (page 4, lines 44-50).

--CD68 is not marker of phagocytic activity per say but a marker of phagolysosomes and should be stated as such

⇒ **Response:** We have stated as such.

Minor Critiques:

1. Color palette used for tSNE could be optimized for ease of reader. The default Seurat/ggplot Hue color palette becomes difficult when number of clusters increases.

⇒ **Response:** We have compared different color palettes and chose a new version with more saturated colors. To help distinguish different clusters, we also have added the number on each cluster.

2. Graph axes and heatmap legends are often not readable due to size of text and compression of the scale. Please fix.

⇒ **Response:** We have enlarged the size of Graph axes and heatmaps legends in the revised manuscript.

Reviewers' Comments:

Reviewer #1:

Remarks to the Author:

Xu et al. substantially improved their manuscript with this resubmission. They added a great deal of functional data to the manuscript and appropriately responded to all the raised concerns. The addition of a cuprizone-mediated demyelination model builds on the characterization experiments and further provides evidence this is a good model to study human microglia. Further edits to methods and figure legends help clarify concerns.

Some minor comments are listed below for the submission to Nature Communications.

1. Page 6 line 42-43 typo: "triple-staining" should be "tripled-stained"
2. Page 11 line 35 typo: "caculated" should be "calculated"

Reviewer #2:

Remarks to the Author:

I agree with the Bennetts (Nature Neuroscience, 2020) that transplantation systems introducing iPSC-derived microglia into mouse CNS are "incredibly promising". But there remain reservations about the validity of the xenotransplantation strategy. They concern the artificial origin of the human in vitro generated "microglia", and about the xenogeneic tissue milieu, which will communicate some, but certainly not all developmentally and functionally relevant signals to the transplanted cells.

This manuscript has been submitted initially to NATURE NEUROSCIENCE, where it went through a very detailed reviewing process. In response to the referees, the paper has been profoundly revised, and visibly improved.

In the following, I focus on the authors' response to referee #3, in particular concerning relevance to MS.

As a basic criticism, the referees qualified the work as mostly technical, without a functional "discovery". The authors countered this by adding detailed single-cell RNA-seq analyses of transplanted phagocytes over time, and during cuprizone triggered demyelination. Introduction of a CNS autoimmune model (an EAE variant) would not have made much sense considering the xeno-barrier between human microglia and the murine immune system.

The authors responded convincingly to most detailed queries, with one exception. They did not transplant murine microglia as a logical control of mechanical effects.

Reviewer #3:

Remarks to the Author:

It is a well written manuscript addressing questions which relevant and interesting. That being said, there are some concerns regarding novelty, such that there have been multiple publications recently with similar findings validating engraftment of stem cell derived microglia into a CSF overexpression murine model, both from the Blurton-Jones lab (Hasselman et al Neuron as a techniques paper 2019, also McQuade et al 2019 and Abud et al 2017 using similar models) as well as the Jaenish lab (Svoboda et al PNAS 2019). Svoboda included scRNAseq of transplanted human stem cell microglia, and Hasselman included scRNAseq comparison between transplanted microglia in the murine model to those transplanted into a backcrossed CSF overexpression and FAD Alzheimer's disease model. I understand this manuscript was submitted prior to the publication of Hasselman and Svoboda and the work arose independently.

The novelty in this manuscript as compared to the aforementioned recent publications is twofold. The first is the comparison between mouse microglia and human microglia derived from the same engrafted brain. These comparisons have not previously been made and are of novel general interest. Secondly is the multiple sclerosis model which has not previously been investigated using transplanted iPSC derived microglia. However – I found their data in the MS model to be shallow

(Fig 6) and would be significantly strengthened by some sequencing (eg scRNAseq of engrafted microglia) data.

Regarding the specific comments made by the previous reviewer #2, with my comments in red: I agree that this paper has strengths, and that there was careful characterization of cell engraftment in figures 1 and 2. If this data had not been previously published this past year it would be novel and well done.

1. Analysis of transcriptomic data: broadly speaking, the authors have not analysed their sequencing data in a way that compellingly demonstrates whether Xeno MGs closely resemble homeostatic human microglia.

This has been addressed (Fig 4 E-F).

1A. Microglia signature genes: There is variation between mouse and human microglia, but many canonical microglial markers are conserved in human. Instead of focusing on these genes, the authors have overall chosen unusual genes and markers to argue for microglial identity. Many of these genes are expressed in microglia but are not microglia-specific, and some are more highly expressed in human compared to mouse microglia. Either way, these genes are not generally accepted as microglial identity markers, including in the human literature. It is critical to clearly show whether genes that indicate true microglial identity are expressed in Xeno MGs, and to validate these in tissue, which the authors have largely not done. In particular, it is important to 1) show expression of *Sall1* and a panel of other microglial identity markers (for example, *Slc2a5*, *P2ry12*, *Tmem119*, and *Olfml3*), 2) to quantify the percent of Xeno Mgs that express these genes, and 3) present the distribution of expression levels, compared to a control population such as pre-engraftment stem cell derived microglia.

This has been addressed (Fig 4 E-F)

1B. Relatedly, in figure 3E, the authors argue that Xeno Mgs resemble microglia based on expression of a list of genes they claim is the top 30 expressed genes in primary human microglia from the Gosselin...Glass, Science 2017 paper. This list does not appear to be the same list as shown in the Glass paper (figure 1B of Glass paper), but even if it were, the most highly expressed genes in a cell type do not inherently define that cell's identity, and may overlap with many other cell types. In the Glass paper, this list was used to show inter-individual variability in highly expressed genes, not to characterize microglial identity.

This was addressed (Fig 4)

1C. CD74 and SPP1 are not human microglia identity markers. In mouse they characterize white matter-associated subsets of microglia that appear during development (Li...Barres Neuron 2019, Hammond...Stevens Immunity 2019). In human, they are enriched in MS patient microglia (Masuda...Prinz Nature 2019). Either way, they do not specifically mark microglia as opposed other macrophages and so do not attest to xeno Mgs being microglia-like. Furthermore, although mRNA appears to be present in the Masuda paper in homeostatic microglia, virtually no microglia express *Spp1* and very few express CD74 in the healthy human brain by immunostaining. Thus, if anything, high expression of these markers suggests non-homeostatic microglia, which should be clarified, for example by tissue staining to compare to published results, and comparative analysis of expression levels in homeostatic microglia from the Masuda paper to Xeno MGs.

This was addressed – the discrepancy between the RNAseq data (high SPP1) and protein (no SPP1) is unclear but has been previously reported (which they also cite)

1D. To show that the mouse CNS environment induces Xeno MG maturation as claimed, the authors would need to compare a very early timepoint after engraftment to a long term timepoint. This is an important piece of data for a resource paper that may form the basis for experiments in other labs predicated on this claim. Furthermore, aging is not the same as maturation from a fetal-like state to an adult state. Thus, validation of maturation using an aging signature does not show maturation.

This was reasonably addressed.

1E. Relatedly, it is critical to see what stem cell-derived microglia-like cells look like transcriptomically prior to engraftment in the brain. This is related to point 1D, and will again allow readers to understand the effects of the brain environment on Xeno Mgs.

This was addressed

1F. Figure 3F - how are the authors controlling for batch effects? It is not clear how downsampling reads addresses this. It is important to explain methods used to test for, and/or adjust for batch effects, which could partially or fully explain their clustering.

This was addressed

2. Synapse engulfment by Xeno MGs is not validated. The authors show PSD95 positive puncta that appear to be inside Xeno MGs in an image reconstruction. They interpret this to mean that Xeno MGs prune synapses in the rodent brain. The inclusion of functional data is appreciated and really strengthens the paper. The finding that human macrophages can engulf mouse synapses, however, is a major finding. It would be the first validation that human microglia-like cells perform microglia-specific functions in vivo. As such, concluding that Xeno MGs engulf synapses requires further validation of the presented imaging data. For one, staining with a post-synaptic marker does not equate to synapse engulfment, which requires colocalized pre- and post synaptic marker staining. Second, confocal imaging alone is insufficient to prove engulfment. This requires a high resolution imaging technique (for example EM, array tomography, possibly super-resolution imaging) to conclusively show synaptic

puncta are inside microglia.

The authors have done super-resolution imaging to try to address this point. The resulting images (Fig 3) are overall not a quality that is convincing. EM would significantly improve this aspect of the manuscript.

3. The results section frequently editorializes about what presented findings might mean, in a way that is misleading. The manuscript is carefully written and makes many good points, but also strongly asserts what findings may mean in the results section. Usually, the authors do qualify these statements so that technically speaking they are not making inappropriate conclusions and rather are speculating, but this reviewer still found these implied conclusions to read like overstatements. Below are some examples, but overall, these unsubstantiated speculations detract from the text. Many of them could be toned down and included in the discussion.

This was addressed

The remainder of the minor points were addressed.

Point-by-point Response to Reviewers' Comments:

Reviewer #1:

Xu et al. substantially improved their manuscript with this resubmission. They added a great deal of functional data to the manuscript and appropriately responded to all the raised concerns. The addition of a cuprizone-mediated demyelination model builds on the characterization experiments and further provides evidence this is a good model to study human microglia. Further edits to methods and figure legends help clarify concerns.

Some minor comments are listed below for the submission to Nature Communications.

1. Page 6 line 42-43 typo: "triple-staining" should be "tripled-stained"

2. Page 11 line 35 typo: "caculated" should be "calculated"

⇒ **Response:** We thank this reviewer for the favorable comments, and we have corrected the typos.

Reviewer #2:

I agree with the Bennetts (Nature Neuroscience, 2020) that transplantation systems introducing iPSC-derived microglia into mouse CNS are "incredibly promising". But there remain reservations about the validity of the xenotransplantation strategy. They concern the artificial origin of the human in vitro generated "microglia", and about the xenogeneic tissue milieu, which will communicate some, but certainly not all developmentally and functionally relevant signals to the transplanted cells.

This manuscript has been submitted initially to NATURE NEUROSCIENCE, where it went through a very detailed reviewing process. In response to the referees, the paper has been profoundly revised, and visibly improved.

In the following, I focus on the authors' response to referee #3, in particular concerning relevance to MS. As a basic criticism, the referees qualified the work as mostly technical, without a functional "discovery". The authors countered this by adding detailed single-cell RNA-seq analyses of transplanted phagocytes over time, and during cuprizone triggered demyelination. Introduction of a CNS autoimmune model (an EAE variant) would not have made much sense considering the xeno-barrier between human microglia and the murine immune system.

The authors responded convincingly to most detailed queries, with one exception. They did not transplant murine microglia as a logical control of mechanical effects.

⇒ **Response:** We thank this reviewer for the positive comments.

We agree with the reviewer's point that transplantation of murine microglia would be a logical control of mechanical effects, if we were to examine the impact of engrafted cells on the host murine brain cells. However, this is outside the scope of this manuscript, because in this study, our goal was to examine expression patterns and functions of the engrafted human microglia.

Reviewer #3:

It is a well written manuscript addressing questions which relevant and interesting. That being said,

there are some concerns regarding novelty, such that there have been multiple publications recently with similar findings validating engraftment of stem cell derived microglia into a CSF overexpression murine model, both from the Blurton-Jones lab (Hasselmann et al Neuron as a techniques paper 2019, also McQuade et al 2019 and Abud et al 2017 using similar models) as well as the Jaenish lab (Svoboda et al PNAS 2019). Svoboda included scRNAseq of transplanted human stem cell microglia, and Hasselmann included scRNAseq comparison between transplanted microglia in the murine model to those transplanted into a backcrossed CSF overexpression and FAD Alzheimer's disease model. I understand this manuscript was submitted prior to the publication of Hasselmann and Svoboda and the work arose independently.

The novelty in this manuscript as compared to the aforementioned recent publications is twofold. The first is the comparison between mouse microglia and human microglia derived from the same engrafted brain. These comparisons have not previously been made and are of novel general interest. Secondly is the multiple sclerosis model which has not previously been investigated using transplanted iPSC derived microglia. However – I found their data in the MS model to be shallow (Fig 6) and would be significantly strengthened by some sequencing (eg scRNAseq of engrafted microglia) data.

Regarding the specific comments made by the previous reviewer #2, with my comments in red: I agree that this paper has strengths, and that there was careful characterization of cell engraftment in figures 1 and 2. If this data had not been previously published this past year it would be novel and well done.

⇒ Response: We thank this reviewer for the positive comments and critical suggestions. We agree with this reviewer that it is important to perform scRNA-seq to profile the properties of human microglia in the MS model. Future studies are warranted to examine this.

1. Analysis of transcriptomic data: broadly speaking, the authors have not analysed their sequencing data in a way that compellingly demonstrates whether Xeno MGs closely resemble homeostatic human microglia.

This has been addressed (Fig 4 E-F).

1A. Microglia signature genes: There is variation between mouse and human microglia, but many canonical microglial markers are conserved in human. Instead of focusing on these genes, the authors have overall chosen unusual genes and markers to argue for microglial identity. Many of these genes are expressed in microglia but are not microglia-specific, and some are more highly expressed in human compared to mouse microglia. Either way, these genes are not generally accepted as microglial identity markers, including in the human literature. It is critical to clearly show whether genes that indicate true microglial identity are expressed in Xeno MGs, and to validate these in tissue, which the authors have largely not done. In particular, it is important to 1) show expression of Sall1 and a panel of other microglial identity markers (for example, Slc2a5, P2ry12, Tmem119, and Olfml3), 2) to quantify the percent of Xeno Mgs that express these genes, and 3) present the distribution of expression levels, compared to a control population such as pre-engraftment stem cell derived microglia. This has been addressed (Fig 4 E-F)

1B. Relatedly, in figure 3E, the authors argue that Xeno Mgs resemble microglia based on expression of a list of genes they claim is the top 30 expressed genes in primary human microglia from the Gosselin...Glass, Science 2017 paper. This list does not appear to be the same list as shown in the Glass paper (figure 1B of Glass paper), but even if it were, the most highly expressed genes in a cell type do not inherently define that cell's identity, and may overlap with many other cell types. In the Glass paper, this list was used to show inter-individual variability in highly expressed genes, not to characterize microglial

identity.

This was addressed (Fig 4)

1C. CD74 and SPP1 are not human microglia identity markers. In mouse they characterize white matter-associated subsets of microglia that appear during development (Li...Barres Neuron 2019, Hammond...Stevens Immunity 2019). In human, they are enriched in MS patient microglia (Masuda...Prinz Nature 2019). Either way, they do not specifically mark microglia as opposed other macrophages and so do not attest to xeno Mgs being microglia-like. Furthermore, although mRNA appears to be present in the Masuda paper in homeostatic microglia, virtually no microglia express Spp1 and very few express CD74 in the healthy human brain by immunostaining. Thus, if anything, high expression of these markers suggests non-homeostatic microglia, which should be clarified, for example by tissue staining to compare to published results, and comparative analysis of expression levels in homeostatic microglia from the Masuda paper to Xeno MGs.

This was addressed – the discrepancy between the RNAseq data (high SPP1) and protein (no SPP1) is unclear but has been previously reported (which they also cite)

1D. To show that the mouse CNS environment induces Xeno MG maturation as claimed, the authors would need to compare a very early timepoint after engraftment to a long term timepoint. This is an important piece of data for a resource paper that may form the basis for experiments in other labs predicated on this claim. Furthermore, aging is not the same as maturation from a fetal-like state to an adult state. Thus, validation of maturation using an aging signature does not show maturation. This was reasonably addressed.

1E. Relatedly, it is critical to see what stem cell-derived microglia-like cells look like transcriptomically prior to engraftment in the brain. This is related to point 1D, and will again allow readers to understand the effects of the brain environment on Xeno Mgs.

This was addressed

1F. Figure 3F - how are the authors controlling for batch effects? It is not clear how downsampling reads addresses this. It is important to explain methods used to test for, and/or adjust for batch effects, which could partially or fully explain their clustering.

This was addressed

⇒ **Response:** We thank this reviewer for the careful evaluation.

2. Synapse engulfment by Xeno MGs is not validated. The authors show PSD95 positive puncta that appear to be inside Xeno MGs in an image reconstruction. They interpret this to mean that Xeno MGs prune synapses in the rodent brain. The inclusion of functional data is appreciated and really strengthens the paper. The finding that human macrophages can engulf mouse synapses, however, is a major finding. It would be the first validation that human microglia-like cells perform microglia-specific functions in vivo. As such, concluding that Xeno MGs engulf synapses requires further validation of the presented imaging data. For one, staining with a post-synaptic marker does not equate to synapse engulfment, which requires colocalized pre-and post synaptic marker staining. Second, confocal imaging alone is insufficient to prove engulfment. This requires a high resolution imaging technique (for example EM, array tomography, possibly super-resolution imaging) to conclusively show synaptic puncta are inside microglia.

The authors have done super-resolution imaging to try to address this point. The resulting images (Fig 3) are overall not a quality that is convincing. EM would significantly improve this aspect of the manuscript.

⇒ **Response:** In order to better demonstrate the synaptic pruning function of human Xeno MG *in vivo*, we have used Imaris software (Bitplane 9.5, Switzerland) to analyze the Airyscan super-resolution images and generated new 3D-surface rendered images, using a method described previously (Schafer et al., J Vis Exp. 2014. PMID: 24962472.). To visualize the synaptic puncta engulfed by microglia, the mask function of Imaris software was employed to subtract any fluorescence that was not within the microglia. This is a commonly used method for analyzing synaptic pruning by microglia (Schafer et al., Neuron. 2012. PMID: 22632727; Bialas and Stevens, Nat Neurosci. 2013. PMID: 24162655; Filipello et al., Immunity. 2018. PMID: 29752066). As shown in new Figure 3B, the 3D-surface rendered images clearly demonstrate that PSD95⁺ synaptic puncta are within hTMEM119⁺ microglia, and some of the PSD95⁺ synaptic puncta are inside of CD68⁺ lysosome.

We have included the new information in the revised manuscript. Please see Results (page 6, lines 36-38) and Methods (page 15, lines 36-39).

3. The results section frequently editorializes about what presented findings might mean, in a way that is misleading. The manuscript is carefully written and makes many good points, but also strongly asserts what findings may mean in the results section. Usually, the authors do qualify these statements so that technically speaking they are not making inappropriate conclusions and rather are speculating, but this reviewer still found these implied conclusions to read like overstatements. Below are some examples, but overall, these unsubstantiated speculations detract from the text. Many of them could be toned down and included in the discussion.

This was addressed

The remainder of the minor points were addressed.

⇒ **Response:** We thank this reviewer for the careful evaluation.